# A two-scale Complexity Measure for Deep Learning Models

**Massimiliano Datres**[1,2]**, Gian Paolo Leonardi**[1]**, Alessio Figalli**[3]**, David Sutter**[4]

[1] Department of Mathematics, University of Trento, Trento
[2] DSH, Bruno Kessler Fondation, Trento
[3] Department of Mathematics, ETH, Zurich
[4] IBM Quantum, IBM Research Europe, Zurich

## Abstract

We introduce a novel capacity measure 2sED for statistical models based on the effective dimension. The new quantity provably bounds the generalization error under mild assumptions on the model. Furthermore, simulations on standard data sets and popular model architectures show that 2sED correlates well with the training error. For Markovian models, we show how to efficiently approximate 2sED from below through a layerwise iterative approach, which allows us to tackle deep learning models with a large number of parameters. Simulation results suggest that the approximation is good for different prominent models and data sets.

## 1 Introduction

Deep learning models are achieving outstanding performances in solving several complex tasks such as image classification problems, object detection [19, 21] and natural language processing [8]. Over-parametrized regimes make DNNs able to extract valuable information from data [33]. Quite surprisingly DNNs typically exhibit impressive generalization capabilities after training [25, 33] without suffering of the expected overfitting. Finding appropriate complexity measures for deep learning models can help in understanding and quantifying their generalization capabilities. Hereafter we propose some essential features that, ideally, should characterize a complexity measure for parametric models:

(P1) it should provide pre-training information consistent with post-training performances;

(P2) its computation should be more efficient and scalable in comparison with a full training & validation process;

(P3) in the case of a feedforward-type model, it should be "modular", i.e., computable in some iterative fashion[1].

Properties (P1)-(P3) are motivated by the goal of finding an efficient tool for model selection in the context of feedforward (stochastic) neural networks.

Notions of complexity measures have appeared in the context of machine learning, with early studies focusing, e.g., on the complexity of decision tree models [7] and logistic regression models [10, 15, 9]. From the perspective of statistical learning theory, the Vapnik-Chervonenkis dimension, commonly called VC dimension [31], is an established complexity measure defined in term of the largest number of points that can be shattered by a class of functions [13]. This complexity dimension has been used

---

[1]Here we are referring to the typical structure of a feedforward-type model, which is a composition of parametric layer maps.

38th Conference on Neural Information Processing Systems (NeurIPS 2024).

to establish data-independent generalization bounds for statistical models [30, 26].

Other notions of model complexity, specifically designed for deep learning models, have been more recently proposed, with the aim of quantifying the expressivity of a DNN [24, 27, 22, 14]. There is, however, a supported evidence that data-independent generalization bound are not universally effective [17, 4]. For this reason, data-dependent notions of complexity have also been introduced, like the Rademacher complexity and the Gaussian complexity. These notions of complexity evaluate the expected noise-fitting-ability of a function class over all data sets drawn according to an unknown data distribution. By means of such data-dependent complexity measures, one obtains generalization bounds that are considerably better than those involving the VC dimension [4, 26, 30]. In any case, computing or estimating the VC dimension, or the Rademacher complexity, is generally a challenging task, feasible only under strong model constraints. Some tight approximations and bounds have been obtained in some specific cases, however they are not general enough to be applicable to complex models like modern deep neural networks [32, 4, 3]. With the aim to provide more easily computable notions of complexity, other definitions have been considered based on the minimum description length (MDL) and on the notion of stochastic complexity [28, 12, 9].

Other complexity measures of more geometric flavour have been defined in terms of the Fisher information matrix (FIM) associated with the statistical model [29, 5, 2]. The geometric properties of the statistical manifold revealed by the FIM collectively contribute to our understanding of the intricate nature of the model, providing a geometric lens through which complexity can be defined, analyzed and interpreted.

**Our contributions.** In Definition 4.1 we propose a notion of model complexity, called *two-scale effective dimension* (2sED), which is motivated by a metric-specific covering argument. We prove a generalization bound derived from 2sED, see Theorem 5.1, thereby theoretically substantiating its efficacy as a measure of complexity. Furthermore, we propose a modular version of the 2sED, called *lower 2sED*, that is specifically tailored for Markovian models. We finally present numerical simulations based on Monte Carlo approximations of 2sED and lower 2sED for various models and datasets. In particular, the experiments confirm properties (P1), (P2), and (P3) for the lower 2sED, thus promoting it as a potential effective tool for model selection.

## 2 Related works

Recently, [5, 2] propose a notion of *effective dimension*, that is, a box-covering dimension related to the number of "Fisher boxes" of a given size that are needed to cover the parameter space. The size of such boxes represents a "scale" at which the model is analysed. Under suitable regularity assumptions on the statistical model and on the loss functional, the generalization error (i.e., the gap between the population error and the empirical error) can be controlled by an expression involving the effective dimension computed with respect to an explicit scale parameter, that depends on the number of samples defining the empirical error. On the one hand, the generalization bounds proved in [2, 1] require the logarithm of the FIM to be Lipschitz which excludes the case of over-parametrized models [16]. On the other hand, the original definition requires a global computation for the statistical model as a whole, hence it does not satisfy properties (P2) and (P3). Indeed, one of the main challenges in the computation of the effective dimension is to determine the eigenvalues of the Fisher information matrix. Note that, for high-dimensional models, even the storage of the Fisher information becomes impractical, despite sophisticated approximation methods such as K-FAC [23]. We focus to address the issues that affect the previous definition by proposing the 2sED and the lower 2sED for Markovian models.

## 3 Preliminaries

Take $\mathcal{X} \subset \mathbb{R}^{d_{in}}$ and $\mathcal{Y} \subset \mathbb{R}^{d_{out}}$ nonempty Borel sets, and denote by $(X, Y) \in \mathcal{X} \times \mathcal{Y}$ a pair of random vectors with (unknown) joint probability distribution $p = p(x, y)$. Let $(X_1, Y_1), \ldots, (X_N, Y_N)$ be i.i.d. copies of $(X, Y)$. A dataset $\mathbb{D} := \{(x_i, y_i) : i = 1, \ldots, N\}$ is understood as a realization of the $N$ random pairs considered before. A *statistical model* on the sample space $\mathcal{X} \times \mathcal{Y}$ is a collection

$$\mathcal{M}_\Theta(\mathcal{X}, \mathcal{Y}) := \{p_\vartheta : \vartheta \in \Theta\},$$

where $p_\vartheta = p_\vartheta(x, y)$ is a joint probability distribution on $\mathcal{X} \times \mathcal{Y}$ for each $\vartheta \in \Theta$, and $\Theta \subseteq \mathbb{R}^d$ is a bounded domain called *parameter space*. In order to stress the functional relation between the input $x$

and the output $y$, it is customary to assume $p_\vartheta = p_\vartheta(y|x)\,p(x)$ of the form $p_\vartheta(x,y) = p_\vartheta(y|x)\,p(x)$ where $p_\vartheta(y|x)$ is a parametric conditional probability, and $p(x)$ denotes the marginal of the unknown distribution $p(x,y)$ on $\mathcal{X}$. We will also assume that the parameter space $\Theta$ is equipped with a probability measure and we denote as $\mathbb{E}_\vartheta$ the expectation with respect to this measure.

Under suitable regularity conditions, we define the *Fisher information matrix* as:

$$F(\vartheta) := \mathbb{E}_{(x,y)\sim p_\vartheta}\left[(\nabla_\vartheta\, l_\vartheta(x,y)) \otimes (\nabla_\vartheta\, l_\vartheta(x,y))\right], \tag{1}$$

where by $a^{\otimes 2} := a \otimes a$ we mean $a \cdot a^T$ (with the convention that $a$ is a column vector) and $l_\vartheta(x,y) := \log p_\vartheta(x,y)$. In other words, the Fisher information matrix is the expectation of the orthogonal projector onto the direction of the gradient of the log-likelihood, scaled by the squared norm of that gradient. It is a symmetric and positive semidefinite $d \times d$ matrix. Its empirical version is

$$F_N(\vartheta) = \frac{1}{N}\sum_{i=1}^{N}(\nabla_\vartheta\, l_\vartheta(X_i,Y_i))) \otimes (\nabla_\vartheta\, l_\vartheta(X_i,Y_i))\,,$$

for some $(X_1,Y_1),\ldots,(X_N,Y_N) \overset{i.i.d.}{\sim} p_\vartheta$. For each $\vartheta \in \Theta$ and $u,v \in \mathbb{R}^d$, if $F(\vartheta)$ is smooth and positive-definite, then $\langle u,v\rangle_\vartheta := \langle F(\vartheta)u,v\rangle$ defines a Riemannian metric on the parameter space $\Theta$, that from now on will be called *Fisher metric* (we shall adopt the same terminology also when the metric is degenerate). In general, the Fisher metric can be considered as the pull-back of a (possibly degenerate) Riemannian metric on $\mathcal{M}_\Theta(\mathcal{X},\mathcal{Y})$ [22].

We also define the *pointed Fisher norm* of a vector $v \in \mathbb{R}^d$ as $\|v\|_{A(\vartheta)} := \sqrt{\langle A(\vartheta)v,v\rangle}$ and the corresponding balls of radius $\varepsilon > 0$ are referred as *Fisher balls* of radius $\varepsilon$ defined as:

$$B_\varepsilon(\vartheta_0) := \left\{\vartheta \in \Theta:\ \|\vartheta - \vartheta_0\|_{F(\vartheta_0)} < \varepsilon\right\}. \tag{2}$$

Some further terminology must be recalled before discussing the generalization bounds. Given a *loss function* $\mathfrak{L}$, i.e., a continuous function $\mathfrak{L}: [0,+\infty) \times [0,+\infty) \to [0,+\infty)$ such that $\mathfrak{L}(a,b) = 0$ if and only if $a = b$, we define the *population risk*

$$R(\vartheta) := \mathbb{E}_{(x,y)\sim p}[\mathfrak{L}(p_\vartheta(y|x),p(y|x))],$$

and the *empirical risk*

$$R_n(\vartheta) := \frac{1}{n}\sum_{i=1}^{n}\mathfrak{L}(p_\vartheta(Y_i|X_i),p(Y_i|X_i))\,,$$

where $(X_i,Y_i) \overset{i.i.d.}{\sim} p$, $i = 1,\ldots,n$. Then, the *generalization error* is defined as

$$\|R - R_n\|_\infty = \sup_{\vartheta \in \Theta}\ |R(\vartheta) - R_n(\vartheta)|\,. \tag{3}$$

## 4   The two-scale effective dimension

Here we consider a notion of complexity for a statistical model $\mathcal{M}_\Theta$, that depends on the properties of the Fisher metric on the parameter space.

**Definition 4.1.** Given $0 < \varepsilon < 1$ and $0 \le \zeta < 1$, we define the *two-scale effective dimension* (or simply 2sED) as

$$d_\zeta(\varepsilon) = \zeta d + (1-\zeta)\frac{\log \mathbb{E}_\vartheta\left[\det\left(I_d + \varepsilon^{\zeta-1}\hat{F}(\vartheta)^{\frac{1}{2}}\right)\right]}{|\log(\varepsilon^{\zeta-1})|}, \tag{4}$$

where

$$\hat{F}(\vartheta) := \begin{cases} \frac{d}{\mathbb{E}_\vartheta[\mathrm{Tr}\, F(\vartheta)]}F(\vartheta) & \text{if } \mathbb{E}_\vartheta[\mathrm{Tr}\, F(\vartheta)] > 0 \\ 0 & \text{otherwise} \end{cases}$$

is the normalized Fisher information matrix, so that whenever the statistical model is not trivial (i.e. not constant with respect to $\vartheta$) the expectation of the trace of $\hat{F}$ satisfies

$$\mathbb{E}_\vartheta[\mathrm{Tr}\, \hat{F}(\vartheta)] = d\,.$$

Note that $d_\zeta(\varepsilon)$ is the convex combination of the dimension $d$ of the parameter space with a slight variant of the effective dimension studied in [2], which is obtained in the special case $\zeta = 0$.

*Remark* 4.2. The 2sED, as the original effective dimension, is inspired by the box-counting or Minkowsky dimension of the statistical manifold $\mathcal{M}_\Theta$ [5, 2] . Given a metric space $(X, d)$ and a subset $S \subset X$, the box-counting dimension of $S$ is defined as

$$\dim_{box}(S) = \lim_{\varepsilon \to 0} \frac{\log \mathcal{N}_d(\varepsilon)}{|\log \varepsilon|} \,,$$

where $\mathcal{N}_d(\varepsilon)$ is the minimum number of $\varepsilon$-size balls (with respect to the metric $d$) needed to cover $S$, also known as the $\varepsilon$-covering number of $S$. The box-counting dimension quantifies how fast $\mathcal{N}_d(\varepsilon)$ changes as the radius $\varepsilon$ approaches zero. Motivated by this notion of fractal dimension, we can define the effective dimension of a statistical model $\mathcal{M}_\vartheta$ at the scale $\varepsilon$ as

$$\dim_{\mathrm{eff}, \varepsilon}(\mathcal{M}_\Theta) = \frac{\log \mathcal{N}_\vartheta(\varepsilon)}{|\log \varepsilon|} \,, \tag{5}$$

where $\mathcal{N}_\vartheta(\varepsilon)$ is the number of Fisher balls of size $\varepsilon$ (defined in (2)) needed to cover $\Theta$. The 2sED defined in (4) is motivated by an upper bound estimate of (5) which is computable for a given statistical model under reasonable regularity assumptions.

*Remark* 4.3. Two parameters show up in the above definition: a micro-scale $\varepsilon > 0$ and an exponent $\zeta \in [0, 1)$ defining a meso-scale $\delta = \varepsilon^\zeta$. The emergence of two scales is tied to the estimation argument for $\mathcal{N}_\vartheta(\varepsilon)$, which requires weaker assumptions compared to those in [5, 2]. The micro-scale is related to the size $\varepsilon$ of Fisher balls that are used to cover a component of the parameter space, while the meso-scale $\varepsilon^\zeta$ represents the size of the components of a partition of the parameter space, that needs to be fixed in order to localise and adapt the micro-scale covering. This covering argument plays a fundamental role in the proof of the generalization bound (Theorem 5.1). A more formal and detailed discussion of the covering argument is reported in the proof of Lemma B.2 in Appendix B. Note that when $\zeta = 0$ we essentially obtain the effective dimension of [2], up to a slight technical difference due to the presence of the square root of the normalized FIM $\hat{F}$. More generally, the 2sED is a convex combination of the dimension of the parameter space and the effective dimension.

*Remark* 4.4. The effective dimension $d_\zeta(\varepsilon)$ can be shown to converge to $\zeta d + (1 - \zeta)\hat{r}$ as $\varepsilon \to 0$, where $\hat{r} := \max_{\vartheta \in \Theta} \mathrm{rank}(\hat{F}(\vartheta))$, see Proposition A.1. The proof follows the strategy of Remark 1 of [2], and is presented in Appendix A for completeness.

## 5 Generalization bounds

It is known that the Fisher information of a statistical model degenerates asymptotically with the number of parameters [16]. This suggests that, in the case of a large (over-parametrized) model, like a deep neural network with high-dimensional layers, the corresponding Fisher information matrix $F(\vartheta)$ should have a lot of small (or possibly zero) eigenvalues. For this reason, in Theorem 5.1 below we will not require the Lipschitz regularity of $\log(F(\vartheta))$, as done in Theorem 1 of [2], because this assumption would imply uniform positive lower bounds on the eigenvalue of $F(\vartheta)$. Without loss of generality, we directly assume $F = \hat{F}$ and $\Theta = [0, 1]^d$, as this can be enforced by a suitable scaling of the model.

We list below a set of hypotheses, that will be required in the generalization bounds:

(i) the map $\vartheta \mapsto p_\vartheta(y|x)$ is of class $C^{1,1}$ uniformly in $(x, y)$;

(ii) there exist two constants $0 < \alpha_1 \le \alpha_2$ such that

$$\alpha_1 \le p(x, y), \ p_\vartheta(x, y) \le \alpha_2$$

for all $x \in \mathcal{X}, y \in \mathcal{Y}, \vartheta \in \Theta$;

(iii) the Fisher matrix field $F(\vartheta)$ is $L$-Lipschitz with respect to the Frobenius norm;

(iv) the loss function $\mathcal{L}$ is bounded by $2b$ and is $\Lambda$-Lipschitz, for some $b, \Lambda > 0$;

(v) the meso-scale parameter $\zeta$ satisfies $\zeta \in [\frac{2}{3}, 1)$.

Some comments about the previous properties are in order. First, property (i) is a mild regularity assumption on the model. Property (ii) prevents degeneration of both probability densities $p(x, y)$ and $p_\vartheta(x, y)$. The $L$-Lipschitz property (iii) is crucial to compare the pointed Fisher norm computed in different $\vartheta \in \Theta$ and it is satisfied for instance by models of class $C^{1,1}$ (but possibly also by more general models). Lipschitz regularity and boundedness of the loss function $\mathcal{L}$ (property (iv)) are standard assumptions (see, e.g., [20]). Finally, property (v) is structurally needed in the proof of the generalization bound (Theorem 5.1) and it is strictly related to the covering argument discussed Lemma B.2 in Appendix B.

**Theorem 5.1.** *Let us assume (i)–(v). Then, there exist explicit constants[2] $C, H, K, n_0 > 0$ such that for any $\gamma \in (0, 1]$, $n \geq n_0$, and $\varepsilon_n = \left(\frac{\log n}{\gamma n}\right)^{3/8}$, we obtain*

$$\mathbf{P}\left(\sup_{\vartheta \in \Theta} |R(\vartheta) - R_n(\vartheta)| \geq C\varepsilon_n\right) \leq H\varepsilon_n^{-d_\zeta(\varepsilon_n)} n^{-\frac{K}{\gamma}}. \tag{6}$$

The proof of Theorem 5.1 is given in Appendix B.

*Remark* 5.2. Under the assumption that the eigenvalues of $F$ are smaller than $\mu$ for some fixed $\mu > 0$, the above result implies the existence of $\gamma_0 > 0$ such that, for $0 < \gamma < \gamma_0$, the right-hand side of (6) vanishes as $n \to \infty$. The upper bound $\gamma_0$ is explicit and depends only on the dimension $d$ and on the properties of the model, see (34). By choosing $\gamma$ as above, the right-hand side of (6) gives an explicit upper bound of the generalization error, that is non-vacuous also for finite $n$ (even though this can be granted only in the under-parametrized regime, i.e. for $n$ large enough).

## 6 The effective dimension of a Markovian model

Markovian models are a family of probabilistic models characterized by a sequential, feed-forward-type structure, see the Markovian property stated below.

Let us consider an integer $L \geq 2$, a probability space $(\Omega, \mathcal{F}, \mathbb{P})$, and a random vector $X_j : \Omega \to \mathcal{X}_j$ for $j = 0, \ldots, L$. Given a parameter space $\Theta = \Theta_1 \times \cdots \times \Theta_L$, a parametric statistical model $\mathcal{M}_\Theta(\mathcal{X}_0, \ldots, \mathcal{X}_L)$ satisfies the Markovian property if and only if for each $p_\vartheta(x_0, \ldots, x_L) \in \mathcal{M}_\Theta(\mathcal{X}_0, \ldots, \mathcal{X}_L)$ and for each $\vartheta = (\vartheta_1, \ldots, \vartheta_L) \in \Theta = \Theta_1 \times \cdots \times \Theta_L$ we have:

$$p_\vartheta(x_0, \ldots, x_L) = p(x_0)p_{\vartheta_1}(x_1|x_0) \cdots p_{\vartheta_L}(x_L|x_{L-1}) \tag{7}$$

where $\vartheta_1, \ldots, \vartheta_L$ are the parameters associated to the model's distribution of $X_1, \ldots, X_L$ respectively. Many well-known and commonly used neural network architectures, such as feed-forward neural networks, can be interpreted as Markovian models with concentrated, Dirac-type probability distributions. A specific evaluation of the effective dimension of these models seems therefore particularly interesting. Exploiting the Markovian property, for $j = 1, \ldots, L$ we define:

$$F(\vartheta_j|x_{j-1}) := \int_{\mathcal{X}_j} (\nabla_{\vartheta_j} l_{\vartheta_j}(x_j|x_{j-1}))^{\otimes 2} p_{\vartheta_j}(dx_j|x_{j-1})$$

where $l_{\vartheta_j}(x_j|x_{j-1}) := \log p_{\vartheta_j}(x_j|x_{j-1})$ and

$$F_j = F_j(\vartheta_1, \ldots, \vartheta_j) := \mathbb{E}_{x_0}\mathbb{E}_{x_1|x_0} \cdots \mathbb{E}_{x_{j-1}|x_{j-2}}[F(\vartheta_j|x_{j-1})],$$

where $\mathbb{E}_{x_0}$ and $\mathbb{E}_{x_j|x_{j-1}}$ denote the (conditional) expectations with respect to $p(x_0)$ and $p_{\vartheta_j}(x_j|x_{j-1})$, respectively. Clearly $F_j$ is a symmetric and positive semidefinite $d_j \times d_j$ matrix (where $d_j$ is the dimension of $\Theta_j$) and represents the $j$-th block of the Fisher information matrix

$$F(\vartheta) = \begin{pmatrix} F_1(\vartheta_1) & 0 & \cdots & 0 \\ 0 & F_2(\vartheta_1, \vartheta_2) & & \vdots \\ \vdots & & \ddots & \vdots \\ 0 & \cdots & \cdots & F_L(\vartheta_1, \ldots, \vartheta_L) \end{pmatrix}. \tag{8}$$

---

[2]the constants can be computed/estimated in terms of the assumptions.

We recall that the two-scale effective dimension (2sED) is

$$d_\zeta(\varepsilon) = \zeta d + \frac{1-\zeta}{|\log \varepsilon|} \log \fint_{\Theta_1} \cdots \fint_{\Theta_L} \prod_{j=1}^{L} det_j \, d\vartheta_1 \cdots d\vartheta_m \,, \tag{9}$$

where $det_j = det_j(\vartheta_1, \ldots, \vartheta_j) := \det\left(I_j + \varepsilon^{-1} F_j^{\frac{1}{2}}\right)$ and $I_j$ denotes the $d_j \times d_j$ identity matrix. Since $F_j$ depends on all the parameters of the previous blocks, it is not possible to directly factorize the multiple integral in (9). Nevertheless, one obtains a more easily computable lower bound of $d_\zeta(\varepsilon)$, called *lower 2sED*, by a single application of Jensen's inequality as hereafter described. Let $d_\zeta^m(\varepsilon)$ be the 2sED associated with the composition of the first $m$ layers, $m \geq 2$. Then:

$$d_\zeta^m(\varepsilon) - d_\zeta^{m-1}(\varepsilon) = \frac{1-\zeta}{|\log \varepsilon|} \log \left( \fint_{\hat{\Theta}_m} \fint_{\Theta_m} det_m \, d\vartheta_m \, d\Phi_m \right)$$

$$\geq \frac{1-\zeta}{|\log \varepsilon|} \fint_{\hat{\Theta}_m} \fint_{\Theta_m} \log det_m \, d\vartheta_m d\Phi_m \,,$$

where we have set $\hat{\Theta}_m := \Theta_1 \times \cdots \times \Theta_{m-1}$ and

$$d\Phi_m = \Phi_m(d\vartheta_1, \ldots, d\vartheta_{m-1}) := \frac{1}{\prod_{j=1}^{m-1} |\Theta_j|} \prod_{j=1}^{m-1} det_j \, d\vartheta_1 \cdots d\vartheta_{m-1} \,.$$

Now, a lower bound of $d_\zeta^m(\varepsilon)$ can be iteratively defined for $m = 1, \ldots, L$ as follows:

$$\underline{d}_\zeta^1(\varepsilon) = \zeta d + \frac{1-\zeta}{|\log \varepsilon|} \log \fint_{\Theta_1} \det(I_1 + \varepsilon^{-1} F_1(\vartheta_1)^{\frac{1}{2}}) \, d\vartheta_1 = \zeta d + \frac{1-\zeta}{|\log \varepsilon|} \log \fint_{\Theta_1} det_1 \, d\vartheta_1$$

$$\vdots \tag{10}$$

$$\underline{d}_\zeta^m(\varepsilon) = \underline{d}_\zeta^{m-1}(\varepsilon) + \frac{1-\zeta}{|\log \varepsilon|} \fint_{\hat{\Theta}_m} \fint_{\Theta_m} \log det_m \, d\vartheta_m \, d\Phi_m \,.$$

From now on we set $\underline{d}_\zeta = \underline{d}_\zeta^L$ and call it the *lower effective dimension* of the Markovian model $\mathcal{M}_\Theta$.

## 7 Experiments

In this section, we present experimental evidence that the behavior of the loss in the training of given parametric models is related both with 2sED (4) and the lower 2sED (10). We compute $\underline{d}_\zeta$ and $d_\zeta$ of different feed-forward neural networks (FNN) such as convolutional neural networks (CNN) and multi-layer perceptrons (MLP). The experiments rely on the computation of the exact eigenvalues via the *numpy.linalg.eig* function. As discussed in Section 8, an interesting research direction (which goes beyond the goal of this paper) would be to investigate strategies to efficiently compute the (lower) 2sED via a suitable approximation of the spectrum of $F$. The feed-forward neural network choice is justified by their architecture characterized by a Markovian dependency structure. Indeed, the flow of information in FNN is unidirectional from input to output, making them representable with a finite acyclic graph. We evaluate $\underline{d}_\zeta$ and $d_\zeta$ on real-world datasets, including Covertype dataset [6], MNIST dataset [11], and CIFAR10 [18]. All simulations are conducted on a 12th Gen Intel(R) Core(TM) i9-12900KF equipped by a NVIDIA GeForce RTX 4090. In the worst-case scenario, the experiments required a maximum RAM memory consumption of 17 GB.

**Description of the models.** To simplify notation and enhance readability, we denote with "MLP $N_0$-$N_1$-$\ldots$-$N_n$" a MLP with $n$ linear layers, each with a width of $N_i$ for $i = 0, \ldots, N$, followed by ReLU activation functions on all layers except the final layer $n$. If we denote with $W^i \in R^{N_i \times N_{i-1}}$ the parameters of the $i$-th layer, we can describe "MLP $N_0$-$N_1$-$\ldots$-$N_n$" through $n$ blocks of operations defined as $O_i(\cdot) = ReLU(W^i \cdot)$ for $i = 1, \ldots, n$. Similarly, "CNN $N_0$-$N_1$-$\ldots$-$N_{n_1}|L_1$-$\ldots$-$L_{n_2}$" refers to a convolutional neural network with $n_1$ convolutional blocks each one of kernel size $N_i$ for $i = 1, \ldots, N_1$ followed by a flattening layer and $n_2$ MLP blocks of width $L_i$ for $i = 1, \ldots, n_2$. Within each convolutional block, the operations of convolution, batch normalization, ReLU activation, and max pooling are performed sequentially. Moreover, the flattening operation is executed by

applying a common convolutional kernel to all the channels of the last convolutional layer. Hence, given the input $A \in \mathbb{R}^{N_c \times k \times k}$ the flattening operation $Flat_K : \mathbb{R}^{N_c \times k \times k} \to \mathbb{R}^{N_c}$ is defined as follows:

$$[Flat_K(A)]_l := A_{l::} \star K = \sum_{i=1}^{k} \sum_{j=1}^{k} A_{l,i,j} K_{i,j},$$

for $l = 1, \ldots, N_c$ where $A_{l::}$ denotes the $\mathbb{R}^{k \times k}$ obtained by fixing the first dimension at index $l$ and $K$ is a $k \times k$ convolutional kernel which is a parametric matrix in applications. This approach effectively reduce the number of parameters allowing us to compute the effective dimension in reasonable time.

**Introducing stochasticity.** In applications, the core architectures of many deep learning models is deterministic, and the stochasticity is usually introduced in the training pipeline rather than in the model itself. This makes deep learning models, like MLPs and CNNs, incompatible with our setting. Therefore, we approximate deterministic feed-forward neural networks with stochastic variants, where the output of each block is Gaussian with mean the current block deterministic output and a small fixed variance $\sigma^2$. In other words, if $N$ is the number of blocks, the output of the $i$-th block $O_i^\sigma$ is given by

$$O_i^\sigma = O_i + \nu \sim \mathcal{N}(O_i, \sigma^2 I),$$

where $O_i$ is the deterministic output of the $i$-th block, $\nu \sim \mathcal{N}(0, \sigma^2)$. For all the subsequent experiments, we will specifically focus on the computation of 2sED and lower-2sED for $\zeta = 0$ and consider the empirical Fisher information matrix $\hat{F}_N$. Note we compare different network topologies while keeping more or less the same number of parameters. Hence, the informative part of the definition of the (lower) 2sED automatically becomes the log ratio.

**Sharpness of lower 2sED.** To empirically validate the lower 2sED, we compute $\underline{d}_0$ and $d_0$ for different stochastic perturbations of feed-forward neural networks, also varying the covering radius $\varepsilon$. We keep constant both the 100 samples used to estimate $\hat{F}_N$ and the 100 vectors of parameters employed for estimating the integrals appearing in (4) and (10). The results are visualized in Figure 1a and Figure 1b.

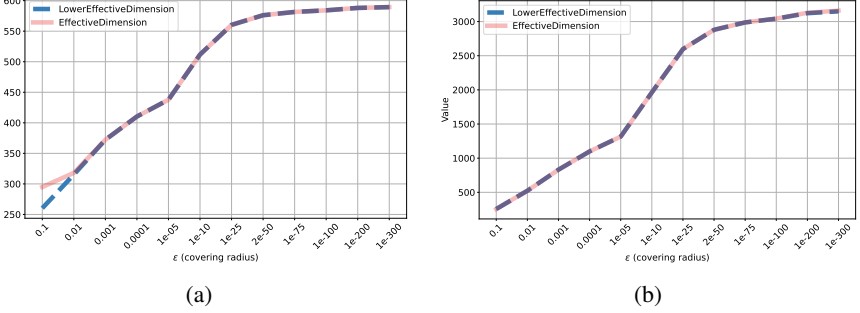

(a)                                                (b)

Figure 1: (a) Difference between 2sED and lower 2sED can not be appreciated, which means that the second is a tight lower bound of the first. Here, the lower 2sED and 2sED of $MLP$ 54-16-7 using 100 Covertype samples and 100 vectors of parameters for the Monte Carlo estimation of $F_N$ is shown; (b)Lower 2sED and 2sED of $CNN$ 7-5|10-50-34-10 using 100 MNIST samples and 100 vectors of parameters for the Monte Carlo estimation of $F_N$ are extremely close.

Notably, the lower bound is sharp in both the MLP and the CNN case suggesting the conclusions regarding model complexity obtained using $\underline{d}_\zeta(\varepsilon)$ are equivalent to those obtained when considering $d_\zeta(\varepsilon)$ for all covering radius $\varepsilon$. It is also worth noticing that lower 2sED exhibits a sequential form, reducing the computational demands when investigating how the model's complexity changes by modifying only its final components.

**Dependence on $\sigma^2$.** We study now the impact of variance $\sigma^2$ on 2sED and lower 2sED. We vary the values of $\sigma^2$ while computing the $\underline{d}_\zeta$ and $d_\zeta$ for different models on Covertype and MNIST dataset. Figure 4 and Figure 5 show that the impact of $\sigma^2$ on $\underline{d}_\zeta$ and $d_\zeta$ is negligible.

**Stability of Monte Carlo estimates.** Monte Carlo integration is crucial in the estimation of both the Fisher information matrix and the integral within $\Theta$ appearing in (4). To ensure the reliability of our

results, we conduct a robustness analysis of the lower 2sED with respect to variations in the number of samples and parameterizations employed for integrals estimation. Figures 2a, 2b, 2c, 2d confirm the stability of the lower 2sED plots with respect to the number of points used in the Monte Carlo approximation. The 2sED is computed for three different models on Covertype, MNIST and Cifar10 dataset.

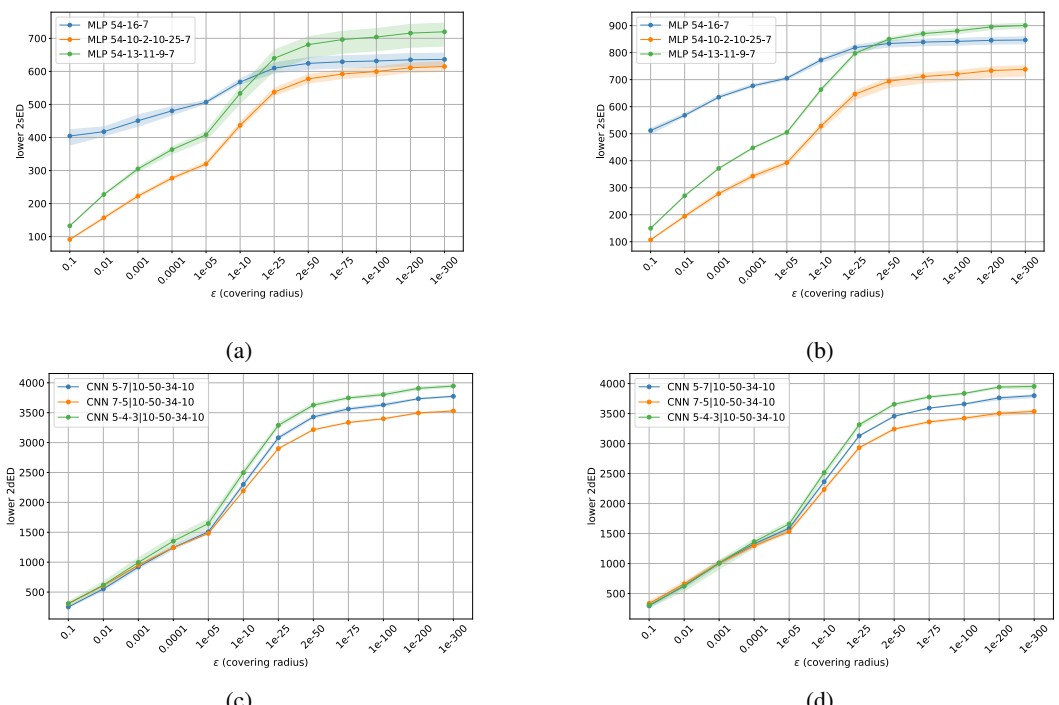

Figure 2: (a) Stability of the lower 2sED of CNNs estimated using 100 Covertype samples and 100 vectors of parameters for the Monte Carlo approximation with the corresponding error margin; (b) Stability of the lower 2sED of MLPs estimated using 1000 Covertype samples and 1000 vectors of parameters for the Monte Carlo approximation with the corresponding error margin; (c) Stability of the lower 2sED of CNNs estimated using 100 Covertype samples and 100 vectors of parameters for the Monte Carlo approximation with the corresponding error margin; (d) Stability of the lower 2sED of CNNs estimated using 500 Covertype samples and 100 vectors of parameters for the Monte Carlo approximation with the corresponding error margin.

**Lower 2sED and training curves.** Finally, we test the relationship between the lower 2sED and the loss minimization. We expect that models with higher values of the lower 2sED can achieve higher accuracy after training. Furthermore, it is crucial to gain a deeper understanding of the role played by the covering radius, denoted as $\varepsilon$, in the 2sED definition. We compute the lower 2sED for three different models with similar dimension on CIFAR10 and Covertype dataset. The dimension of these models is reported in Table 1. MLP 54-10-2-10-25-7 is characterized by a bottleneck structure in the middle of its architecture. A loss of information due to this bottleneck is therefore expected as data are mapped into a significantly lower dimensional space. Consequently, the expressiveness of this model is expected to be lower compared to the other two models, even though it is bigger than MLP 54-16-7 in terms of number of parameters. In Figure 3a, this expected behaviour is effectively captured by the lower 2sE, as indicated by the lower position of the red curve in comparison to the other two curves.

Furthermore, the position of the curves change varying the covering radius $\varepsilon$. The MLP 54-16-7 model exhibits greater expressiveness within this range of $\epsilon$. Conversely, for smaller values of $\epsilon$, MLP 54-13-11-9-7 appears to be more expressive. This behaviour is empirically validated by the experiments. In Figure 3c and Figure 3e we observe the training loss curve for the three models when trained with only 10000 and 100000 data respectively. Note that MLP 54-16-7 is the one achieving the lower training loss using 10000 data, while MLP 54-13-11-9-7 is consistently better

Table 1: Number of model's parameters

| MODEL | NUMBER OF PARAMETERS |
|---|---|
| MLP 54-16-7 | 976 |
| MLP 54-13-11-9-7 | 1007 |
| MLP 54-10-2-10-25-7 | 1005 |
| CNN 7-5|10-50-34-10 | 4493 |
| CNN 3-5-3-6|10-50-34-10 | 10034 |
| CNN 3-6-5-3|10-50-34-10 | 10041 |

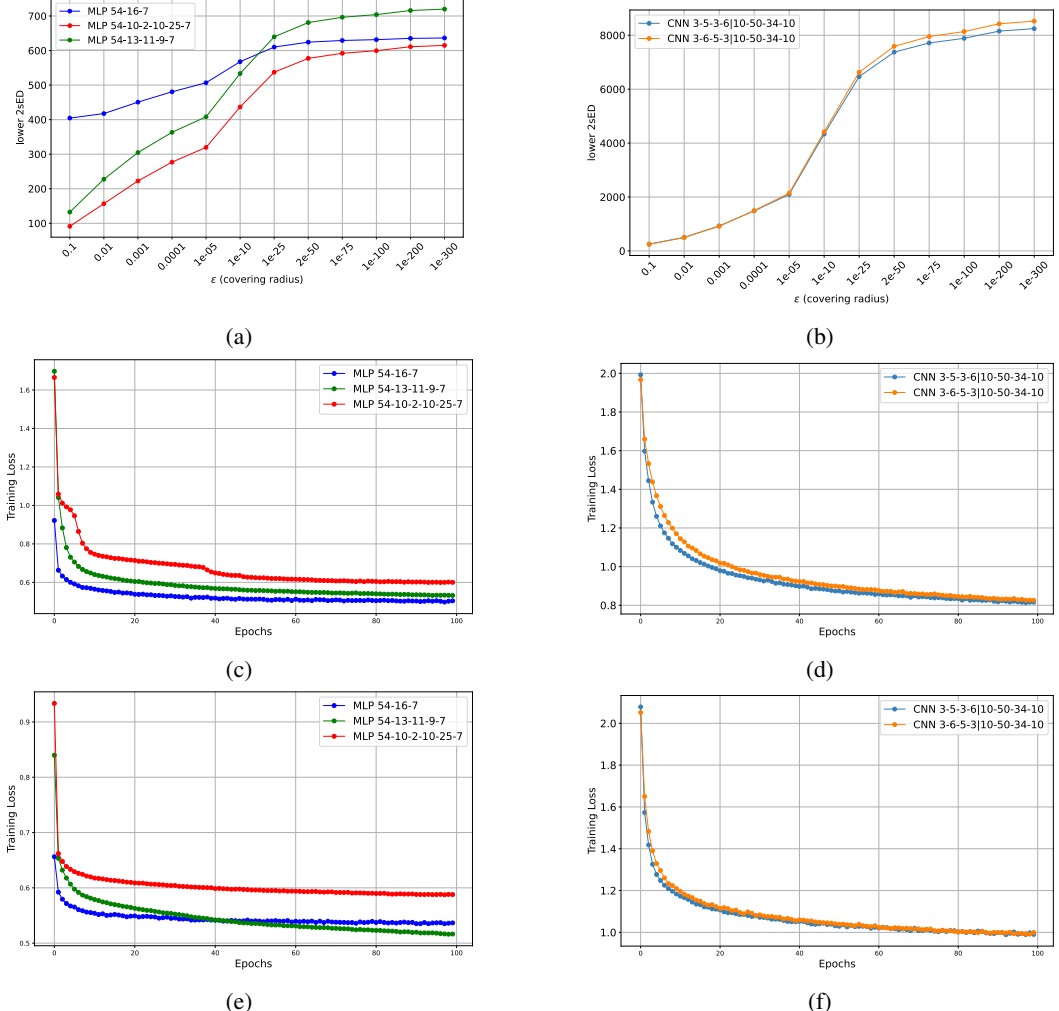

Figure 3: (a) Estimated lower 2sED of three different MLP architectures using 100 Covertype samples and 100 different vectors of parameters for the Monte Carlo estimation of $F_N$; (b) Estimated lower 2sED of three different CNN architectures using 100 CIFAR10 samples and 100 different vectors of parameters for the Monte Carlo estimation of $F_N$; (c) Training loss plots of MLPs on 10000 random Covertype samples using Adam with learning rate $1e^{-3}$ and a batch size 64; (d) Training loss plots of CNNs on CIFAR10 using Adam optimizer with learning rate $1e^{-3}$ and a batch size 512;(e) Training loss plot of MLPs on 100000 Covertype samples using Adam optimizer with learning rate $1e^{-3}$ and a batch size 64; (f) Training loss plots of CNNs on augmented CIFAR10 (double the original size) using Adam optimizer with learning rate $1e^{-3}$ and a batch size 512.

with 100000 data. The empirical correlation between training losses and lower 2sED underscores the capacity of 2sED as a reliable measure for describing the training capabilities of neural networks. We conducted additional experiments, manipulating the number of training data points. The outcomes align consistently with the previously described results. This further confirms its effectiveness as a capacity metric.

**Other experiments.** Other experiments in this direction are performed on the CIFAR10 and MNIST dataset and the results are reported in Figures 3b, 3d, 3f, 6, 7, 8, 9 varying batch sizes providing additional empirical evidence that supports our findings.

## 8    Conclusions

We propose a novel complexity measure called 2sED strongly related with the geometric properties of the statistical manifold. This notion allows to bound the generalization error under mild assumptions on the model (see Theorem 5.1) to theoretically justify the 2sED as a complexity measure. At the same time, a modular version of the 2sED, called lower 2sED, specifically tailored for Markovian models, is introduced as a tight lower bound for the 2sED. The lower 2sED can be computed sequentially layer-by-layer, which drastically reduces the computational effort and the storage consumption compared to the original 2sED. Consequently, the lower 2sED can be computed for more complex (deeper) models than those considered in previous works.

Finally, numerical simulations based on Monte Carlo approximation of the 2sED and the lower 2sED for various models and datasets are presented. The experiments remarkably confirm desirable properties (P1), (P2), and (P3) for the lower 2sED. These experiments show that the lower 2sED represents a tight approximation of the 2sED. The resulting relation between the scale parameter, the number of training data and the training error suggests that the lower 2sED is a reliable complexity measure that can be used as a tool for training-free model selection. Indeed, it can be accurately estimated for general models, distinguishing it from other complexity measures that are often challenging to compute directly and may only be estimated, often assuming a precise model structure.

**Limitations and Future perspectives.** We study a new notion of complexity for deep learning models and we show empirical evidence that the lower 2sED effectively captures the training behaviour. However, the computation of the lower 2sED for large-scale machine learning models is complicated by the dimension of the FIM, whose eigenvalue problem is computationally intractable. In light of the theoretical nature of this work, the implementation of the 2sED has not been optimized. Exploring code optimization would be an interesting direction for future research and necessary to study the lower 2sED of bigger models.

Therefore, an interesting research direction is to develop techniques for further reducing the computational cost of the lower 2sED. The main goal is to approximate the eigenvalue distribution of the FIM, rather than computing exactly each eigenvalue. This would consistently improve the effectiveness of the lower 2sED as a model selection criterion, e.g., in the step-by-step design of deep feedforward neural networks. Another avenue for future research is to design variants of 2sED and lower 2sED specifically adapted to very large neural networks, which characterize modern deep learning architectures. This not only would bring out a deeper understanding of the complexity and the generalization capabilities of huge machine learning models, but it could also provide more efficient approximations of the (lower) 2sED for them.

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

# A  Asymptotic property of $d_\zeta(\varepsilon)$

In this section, we prove the following result.

**Proposition A.1.** *Let $\hat{r}$ denote the maximum rank of the Fisher matrix $\hat{F}(\vartheta)$ and let $\mu > 0$ be an upper bound for all the eigenvalues of $\hat{F}(\vartheta)$, for all $\vartheta \in \Theta$. Then, for all $\zeta \in [0,1)$ and $0 < \varepsilon < 1$ we have*

$$d_\zeta(\varepsilon) \le \zeta d + \hat{r}\left(1 - \zeta + \frac{\log(1 + \mu^{1/2})}{|\log \varepsilon|}\right) \tag{11}$$

*and, moreover,*

$$\lim_{\varepsilon \to 0} d_\zeta(\varepsilon) = \zeta d + (1 - \zeta)\hat{r}\,.$$

*Proof.* Let us fix $\zeta, \varepsilon$ as above. Denoting by $r_\vartheta$ the rank of $\hat{F}(\vartheta)$, we have:

$$\begin{aligned}
d_\zeta(\varepsilon) &= \zeta d + \frac{\log \fint_\Theta \det(Id_d + \varepsilon^{\zeta - 1}\hat{F}^{1/2}(\vartheta))\,d\vartheta}{|\log \varepsilon|} \\
&= \zeta d + \frac{\log \fint_\Theta \prod_{i=1}^{r_\vartheta}(1 + \varepsilon^{\zeta - 1}\lambda_i^{1/2}(\vartheta))\,d\vartheta}{|\log \varepsilon|} \\
&\le \zeta d + \log \frac{\fint_\Theta \varepsilon^{(\zeta - 1)r_\vartheta}\prod_{i=1}^{r_\vartheta}(1 + \lambda_i^{1/2}(\vartheta))\,d\vartheta}{|\log \varepsilon|} \\
&\le \zeta d + \frac{\log \varepsilon^{(\zeta - 1)\hat{r}}}{|\log \varepsilon|} + \frac{\log \fint_\Theta \prod_{i=1}^{r_\vartheta}(1 + \lambda_i^{1/2}(\vartheta))\,d\vartheta}{|\log \varepsilon|}\,,
\end{aligned}$$

where $\lambda_i(\vartheta)$ are the nonzero eigenvalues of $\hat{F}(\vartheta)$. Notice that $\log \fint_\Theta \prod_{i=1}^{r_\vartheta}(1 + \lambda_i^{1/2}(\vartheta))\,d\vartheta$ is finite. Indeed, $0 \le \lambda_i(\vartheta) \le \mu$ by assumption. Then it holds

$$1 \le \prod_{i=1}^{r_\vartheta}(1 + \lambda_i^{1/2}(\vartheta)) \le (1 + \mu^{1/2})^{\hat{r}}\,.$$

This implies that

$$1 \le \fint_\Theta \prod_{i=1}^{r_\vartheta}(1 + \lambda_i^{1/2}(\vartheta)) \le (1 + \mu^{1/2})^{\hat{r}}$$

and therefore (11). We thus conclude that

$$\lim_{\varepsilon \to 0} d_\zeta(\varepsilon) \le \zeta d + (1 - \zeta)\hat{r}\,. \tag{12}$$

To see the other inequality, let us consider $\mathcal{A} := \{\vartheta \in \Theta : r_\vartheta = \hat{r}\}$. Notice that $\mathcal{A} \subset \Theta$ and hence $|\mathcal{A}| < \infty$. Also, by continuity of the Fisher matrix, the set $\mathcal{A}$ has positive measure. Then, we have

$$\begin{aligned}
d_\zeta(\varepsilon) &= \zeta d + \frac{\log \fint_\Theta \det(Id_d + \varepsilon^{\zeta - 1}\hat{F}^{\frac{1}{2}}(\vartheta))\,d\vartheta}{|\log \varepsilon|} \\
&\ge \zeta d + \frac{\log \fint_\mathcal{A} \det(Id_d + \varepsilon^{\zeta - 1}\hat{F}^{\frac{1}{2}}(\vartheta))\,d\vartheta}{|\log \varepsilon|} \\
&= \zeta d + \frac{\log \fint_\mathcal{A} \det(Id_{d_\vartheta} + \varepsilon^{\zeta - 1}\hat{F}_0^{\frac{1}{2}}(\vartheta))\,d\vartheta}{|\log \varepsilon|}\,,
\end{aligned}$$

where $d_\vartheta$ is the number of non-zero eigenvalues of $\hat{F}(\vartheta)$ and $\hat{F}_0(\vartheta)$ is the diagonal $d_\vartheta \times d_\vartheta$ containing only the $d_\vartheta$ non-zero eigenvalues of $\hat{F}(\vartheta)$ for all $\vartheta \in \Theta$. This yields

$$
\begin{aligned}
d_\zeta(\varepsilon) &\geq \zeta d + \frac{\log \fint_{\mathcal{A}} \det(Id_d) + \det(\varepsilon^{\zeta-1}\hat{F}_0^{\frac{1}{2}}(\vartheta))\, d\vartheta}{|\log \varepsilon|} \\
&= \zeta d + \frac{\log |\mathcal{A}|}{|\log \varepsilon|} + \frac{\log \fint_{\mathcal{A}} \prod_{i=1}^{\hat{r}} \varepsilon^{\zeta-1} \lambda_i^{\frac{1}{2}}(\vartheta)\, d\vartheta}{|\log \varepsilon|} \\
&= \zeta d + \frac{\log |\mathcal{A}|}{|\log \varepsilon|} + \frac{\log \fint_{\mathcal{A}} \prod_{i=1}^{\hat{r}} \varepsilon^{\zeta-1} \lambda_i^{\frac{1}{2}}(\vartheta)\, d\vartheta}{|\log \varepsilon|} \\
&= \zeta d + \hat{r}(\zeta - 1) \frac{\log \varepsilon}{|\log \varepsilon|} + \frac{\log \fint_{\mathcal{A}} \lambda_i^{\frac{1}{2}}(\vartheta)\, d\vartheta}{|\log \varepsilon|} \\
&= \zeta d + \hat{r}(1 - \zeta) + \frac{\log \fint_{\mathcal{A}} \lambda_i^{\frac{1}{2}}(\vartheta)\, d\vartheta}{|\log \varepsilon|}\,.
\end{aligned}
$$

Notice now that since $\lambda_i(\vartheta) \neq 0$ for all $\vartheta \in \Theta$, it holds that $log \fint_{\mathcal{A}} \lambda_i^{\frac{1}{2}}(\vartheta)\, d\vartheta < \infty$ and so:

$$
\lim_{\varepsilon \to 0} \frac{\log \fint_{\mathcal{A}} \lambda_i^{\frac{1}{2}}(\vartheta)\, d\vartheta}{|\log \varepsilon|} = 0\,.
$$

Therefore

$$
\lim_{\varepsilon \to 0} d_\zeta(\varepsilon) \geq \zeta d + \hat{r}(1 - \zeta)\,. \tag{13}
$$

Combining (12) and (13), we conclude

$$
\lim_{\varepsilon \to 0} d_\zeta(\varepsilon) = \zeta d + \hat{r}(1 - \zeta)\,.
$$

$\square$

## B  Proof of the generalization bound

We conveniently introduce additional terminology and a few of new definitions concerning $d \times d$ symmetric matrices and matrix fields. We denote by $S_+^d(\mathbb{R})$ the set of real $d \times d$ symmetric and positive semidefinite matrices.

Recall that, for all $A \in S_+^d(\mathbb{R})$ and $x \in \mathbb{R}^d$,

$$
|\langle Ax, x \rangle| \leq \|A\| \|x\|^2\,,
$$

where $\|\cdot\|$ denotes the Frobenius norm (this is because the Frobenius norm bounds from above the operator norm). Let $\beta > 0$, then for $A \in S_+^d(\mathbb{R})$ we define $A_\beta$ by replacing its eigenvalues smaller than $\beta$ with $\beta$. In practice, we consider a spectral basis $\{u_j\}_{j=1,\ldots d}$ with its corresponding sequence of eigenvalues $\{\lambda_j\}_{j=1,\ldots,d}$, then define

$$
A_\beta = \sum_{j=1}^{d} \max(\lambda_j, \beta) u_j \otimes u_j\,. \tag{14}
$$

Note that this definition does not depend on the choice of the basis because the matrix $A$ only depends on its eigenvalues and eigenspaces. Given a matrix field $A : \Theta \to S_+^d(\mathbb{R})$, we define the pointed $A$ norm of a (tangent) vector $v \in \mathbb{R}^d$ at $\vartheta \in \Theta$ as

$$
\|v\|_{A(\vartheta)} := \sqrt{\langle A(\vartheta)v, v \rangle}\,.
$$

Let $A \in S_+^d(\mathbb{R})$, we choose a spectral basis $U = \{u_i\}_{i=1..d}$ for $A$ and, for any $v \in \mathbb{R}^d$, we set

$$
[v]_{A,U} := \max_{i=1,\ldots,d} \sqrt{\lambda_i} |\langle v, u_i \rangle|\,, \tag{15}
$$

where $\lambda_i$ is the eigenvalue corresponding to the eigenvector $u_i$.

**Lemma B.1.** *Let $A : \Theta \to S_+^d(\mathbb{R})$ be a $L$-Lipschitz matrix field. Then for all $\beta > 0$, $v \in \mathbb{R}^d$, and $\vartheta_1, \vartheta_2 \in \Theta$, one has*

$$\|v\|_{A_\beta(\vartheta_2)}^2 \le \big(3 + \omega_\beta(|\vartheta_1 - \vartheta_2|)\big)\|v\|_{A_\beta(\vartheta_1)}^2 \,, \tag{16}$$

*where*

$$\omega_\beta(t) = L\beta^{-1}t \,. \tag{17}$$

*Proof.* We first note that

$$|\langle(A - A_\beta)v, v\rangle| \le \beta|v|^2. \tag{18}$$

To see this we write $v$ in spectral coordinates and compute

$$\langle(A - A_\beta)v, v\rangle = \sum_{j=1}^d (\lambda_j - \max(\lambda_j, \beta))^2 v_j^2 = \sum_{j:\ \lambda_j < \beta} (\lambda_j - \beta)^2 v_j^2 \le \beta^2 |v|^2 \,,$$

whence (18) follows. Now by (18) we obtain the conclusion:

$$\begin{aligned}
\|v\|_{A_\beta(\vartheta_2)}^2 &\le \langle A_\beta(\vartheta_1)v, v\rangle + |\langle(A_\beta(\vartheta_1) - A_\beta(\vartheta_2))v, v\rangle| \\
&\le \langle A_\beta(\vartheta_1)v, v\rangle + \beta|v|^2 + |\langle A(\vartheta_1)v, v\rangle - \langle A(\vartheta_2)v, v\rangle| + \beta|v|^2 \\
&\le \langle A_\beta(\vartheta_1)v, v\rangle + (2 + L\beta^{-1}|\vartheta_1 - \vartheta_2|)\beta|v|^2 \\
&\le (3 + L\beta^{-1}|\vartheta_1 - \vartheta_2|)\|v\|_{A_\beta(\vartheta_1)}^2 \,,
\end{aligned}$$

where in the last inequality we have used the fact that

$$\|v\|_{A_\beta}^2 = \sum_{j=1}^d \max(\lambda_j, \beta)\, v_j^2 \ge \beta|v|^2 \,.$$

$\square$

**Lemma B.2.** *Let $0 < \varepsilon < 1$, $\zeta \in [\frac{2}{3}, 1)$, $\Theta = [0, 1]^d$, and assume that the Fisher matrix field $F(\vartheta)$ is $L$-Lipschitz. Then, $F$ admits an $L$-Lipschitz extension to the whole $\mathbb{R}^d$, and $\Theta$ can be covered by $C_d\, \varepsilon^{-d_\zeta(\varepsilon)}$ Fisher balls of radius $\varepsilon$, where $d_\zeta(\varepsilon)$ is as in (4), and $C_d$ is a dimensional constant.*

*Proof.* The fact that any $L$-Lipschitz mapping from a subset of $\mathbb{R}^d$ into $\mathbb{R}^m$ admits an $L$-Lipschitz extension to the whole $\mathbb{R}^d$ is classically known as Kirszbraun's Theorem. Consider now a partition $\mathcal{Q}$ of $\Theta$ made by closed cubes with mutually disjoint interior and side $\delta = \delta(Q) = \varepsilon^\zeta$, and let $Q$ be one of these cubes. Set

$$\beta = \varepsilon^2/\delta^2 = \varepsilon^{2-2\zeta} \,, \tag{19}$$

fix a generic $\vartheta_Q \in Q$ and a spectral basis $U_Q$ for $F(\vartheta_Q)$. Then, the $\beta$-Fisher box centered in $\vartheta_Q$ of radius $\varepsilon > 0$ is defined as

$$\mathrm{Box}_{\beta,\varepsilon}(\vartheta_Q) := \big\{\vartheta \in \Theta : \ [\vartheta - \vartheta_Q]_{F_\beta(\vartheta_Q), U_Q} < \varepsilon\big\} \,.$$

Let $S_Q$ be the Euclidean ball circumscribed to $Q$. Consider a partition of $\mathbb{R}^d$ by means of translated copies of $\mathrm{Box}_{\beta,\varepsilon}(\vartheta_Q)$, then the minimum number of such boxes that have a nonempty intersection with $Q$ is bounded from above by the number $\tilde{k} = \tilde{k}(Q, \beta, \varepsilon)$ of boxes that have a nonempty intersection with $S_Q$. The volume of each copy of $\mathrm{Box}_{\beta,\varepsilon}(\vartheta_Q)$ is given by

$$|\mathrm{Box}_{\beta,\varepsilon}(\vartheta_Q)| = \prod_{i=1}^d \frac{2\varepsilon}{\sqrt{\lambda_{i,\beta}(\vartheta_Q)}} \,.$$

At the same time, the union of the covering boxes is contained in $S_Q + B_{2\varepsilon\sqrt{d}/\sqrt{\beta}}$, i.e., in a Euclidean ball $B_\rho$ with

$$\rho = \sqrt{d}\left(\frac{\delta}{2} + 2\frac{\varepsilon}{\sqrt{\beta}}\right) = \frac{5}{2}\sqrt{d}\delta \,, \tag{20}$$

hence its volume is bounded from above by $|B_\rho| = \alpha_d\, \rho^d$, where $\alpha_d = \pi^{d/2}/\Gamma(d/2 + 1)$ is the volume of an Euclidean ball of radius 1 in $\mathbb{R}^d$, and $\Gamma(\cdot)$ is Euler's Gamma function. Therefore we

can estimate $\tilde{k}$ from above by the ratio between the upper bound on the volume of the union of the boxes and the volume of a single box. We obtain

$$\tilde{k} \leq \frac{|B_\rho|}{|\text{Box}_{\beta,\varepsilon}(\vartheta_Q)|} = \frac{\alpha_d \left(\sqrt{d}(\delta/2 + 2\varepsilon/\sqrt{\beta})\right)^d}{\prod_{i=1}^d \frac{2\varepsilon}{\sqrt{\lambda_{i,\beta}(\vartheta_Q)}}} = \frac{\alpha_d \left(5/2\sqrt{d}\delta\right)^d}{\prod_{i=1}^d \frac{2\varepsilon}{\sqrt{\lambda_{i,\beta}(\vartheta_Q)}}}$$

$$\leq c_d \prod_{i=1}^d \left\lceil \sqrt{\frac{\lambda_{i,\beta}(\vartheta_Q)}{\beta}} \right\rceil = c_d \prod_{i=1}^d \left\lceil \sqrt{\frac{\lambda_i(\vartheta_Q)}{\beta}} \right\rceil, \tag{21}$$

where we have used the special rounding function

$$\lceil x \rceil = \min\{k \in \mathbb{N} : k \geq \max(x, 1)\}$$

(note that $\lceil x \rceil \geq 1$ for all $x$), and where $c_d = \alpha_d (5/4)^d d^{d/2}$. Note that, by Stirling's formula $\Gamma(x+1) \sim \sqrt{2\pi x}(x/e)^x$ valid as $x \to +\infty$, we deduce that $c_d < (25\pi e/8)^{d/2} < 6^d$ for $d$ large enough.

Let us notice that for all $\vartheta \in S_Q$, the translated copy of $\text{Box}_{\beta,\varepsilon}(\vartheta_Q)$ centered in $\vartheta$ is contained in the corresponding translated copy of $B_{\beta,\varepsilon'}(\vartheta_Q)$ centered in $\vartheta$, where:

$$B_{\beta,\varepsilon'}(\vartheta_Q) := \{\xi \in \Theta : \|\xi - \vartheta_Q\|_{F_\beta(\vartheta_Q)} < \varepsilon'\}$$

and $\varepsilon' = \sqrt{d}\varepsilon$. Indeed, let $\vartheta$ be the center of the translated copy $\widetilde{\text{Box}}$ of $\text{Box}_{\beta,\varepsilon}(\vartheta_Q)$, then for each $\xi \in S_Q \cap \widetilde{\text{Box}}$ one has by definition $[\xi - \vartheta]_{F_\beta(\vartheta_Q),U_Q} < \varepsilon$. Consequently we have

$$\|\xi - \vartheta\|_{F_\beta(\vartheta_Q)} = \sqrt{\sum_{i=1}^d \lambda_{\beta,i}(\vartheta_Q) |\langle \xi - \vartheta, u_i(\vartheta_Q) \rangle|^2}$$

$$\leq \sqrt{d} \max_{i=1,\ldots,d} \sqrt{\lambda_{\beta,i}(\vartheta_Q)} |\langle \xi - \vartheta, u_i(\vartheta_Q) \rangle|$$

$$\leq \sqrt{d}[\xi - \vartheta]_{F_\beta(\vartheta_Q),U_Q}$$

$$\leq \varepsilon'$$

Now we notice that for all $\vartheta \in S_Q$ the translated copy of $B_{\beta,\varepsilon'}(\vartheta_Q)$ centered in $\vartheta$ is contained in $B_{\beta,\varepsilon''}(\vartheta)$, where

$$\varepsilon'' = \varepsilon\sqrt{d(3 + 5\sqrt{d}L)}. \tag{22}$$

Indeed, let $\vartheta$ be the center of the translated copy $\widetilde{B}$ of $B_{\beta,\varepsilon}(\vartheta_Q)$, then for each $\xi \in S_Q \cap \widetilde{B}$ one has by definition $\|\xi - \vartheta\|_{F_\beta(\vartheta_Q)} < \varepsilon'$. Consequently, by Lemma B.1 and by the fact that both $\vartheta$ and $\vartheta_Q$ are contained in $B_R$, one gets

$$\|\xi - \vartheta\|_{F_\beta(\vartheta)} \leq \|\xi - \vartheta\|_{F_\beta(\vartheta_Q)} \sqrt{3 + \omega_\beta(|\vartheta - \vartheta_Q|)}$$

$$\leq \varepsilon'\sqrt{3 + \omega_\beta(|\vartheta - \vartheta_Q|)}$$

$$\leq \varepsilon\sqrt{d(3 + \omega_\beta(5\sqrt{d}\delta))}$$

$$\leq \varepsilon\sqrt{d(3 + 5\sqrt{d}L)}.$$

where in the last step we have used

$$\omega_\beta(5\sqrt{d}\delta) = 5\sqrt{d}L\varepsilon^{3\zeta-2} \leq 5\sqrt{d}L, \tag{23}$$

with the last inequality following from the assumption $\zeta \geq 2/3$.

The previous estimate shows that there exists a covering of $Q$ by means of at most $k_Q$ balls of the form $B_j = B_{\beta, \varepsilon''}(\vartheta_j)$, with $j = 1, \ldots, k_Q$. Therefore, by combining (19), (21) and (22), we get

$$
\begin{aligned}
k_Q \leq \tilde{k} &\leq c_d \prod_{i=1}^{d} \left\lceil \sqrt{\frac{\delta^2 \lambda_i(\vartheta_Q)}{\varepsilon^2}} \right\rceil \\
&\leq c_d \prod_{i=1}^{d} \left\lceil \sqrt{\delta^2 d(3 + 5\sqrt{d}L)\lambda_i(\vartheta_Q)/(\varepsilon'')^2} \right\rceil \\
&\leq c_d (d(3 + 5\sqrt{d}L))^{\frac{d}{2}} |Q| \prod_{i=1}^{d} \left( (\varepsilon'')^{-1}\sqrt{\lambda_i(\vartheta_Q)} + \delta^{-1} \right) \\
&\leq c_d (d(3 + 5\sqrt{d}L))^{\frac{d}{2}} \varepsilon^{-\zeta d} |Q| \prod_{i=1}^{d} \left( \frac{\varepsilon^\zeta}{\varepsilon''}\sqrt{\lambda_i(\vartheta_Q)} + 1 \right) \\
&\leq c_d (d(3 + 5\sqrt{d}L))^{\frac{d}{2}} \left( \frac{\varepsilon''}{d(3 + 5\sqrt{d}L)} \right)^{-\zeta d} |Q| \prod_{i=1}^{d} \left( \frac{(\varepsilon'')^{\zeta-1}}{(d(3 + 5\sqrt{d}L))^\zeta}\sqrt{\lambda_i(\vartheta_Q)} + 1 \right) \\
&\leq c_d (d(3 + 5\sqrt{d}L))^{\frac{d}{2}+\zeta d} (\varepsilon'')^{-\zeta d} |Q| \prod_{i=1}^{d} \left( (\varepsilon'')^{\zeta-1}\sqrt{\lambda_i(\vartheta_Q)} + 1 \right) \\
&\leq c_d (d(3 + 5\sqrt{d}L))^{\frac{3}{2}d} (\varepsilon'')^{-\zeta d} |Q| \prod_{i=1}^{d} \left( (\varepsilon'')^{\zeta-1}\sqrt{\lambda_i(\vartheta_Q)} + 1 \right),
\end{aligned}
$$

where we have used that $\lceil xy \rceil \leq xy + 1$ for all $x, y \geq 0$. Therefore, by conveniently writing $\varepsilon$ instead of $\varepsilon''$, we obtain that the number $k_Q$ of $\beta$-Fisher balls of size $\varepsilon$ that are needed to cover $Q$ satisfies

$$
k_Q \leq C_d \varepsilon^{-\zeta d} |Q| \prod_{i=1}^{d} \left( 1 + \varepsilon^{\zeta-1}\sqrt{\lambda_i(\vartheta_Q)} \right) = C_d \varepsilon^{-\zeta d} |Q| \det \left( I + \varepsilon^{\zeta-1} F(\vartheta_Q)^{\frac{1}{2}} \right),
$$

where now

$$
\beta = \left( \frac{\varepsilon^2}{d(3 + 5L\sqrt{d})} \right)^{1-\zeta} \tag{24}
$$

and $C_d = c_d d^{3d/2}(3 + 5\sqrt{d}L)^{3d/2}$. Finally, if we denote by $k(\varepsilon)$ the cardinality of the least number of $\beta$-Fisher balls of size $\varepsilon$ that are needed to cover $\Theta$, by summing over $Q$ and choosing $\vartheta_Q$ as a minimum point for $\det(I + \varepsilon^{\zeta-1} F(\vartheta)^{\frac{1}{2}})$ when $\vartheta \in Q$, we obtain

$$
k(\varepsilon) \leq C_d \varepsilon^{-\zeta d} \int_\Theta \det \left( I + \varepsilon^{\zeta-1} F(\vartheta)^{\frac{1}{2}} \right) d\vartheta = C_d \varepsilon^{-d_\zeta(\varepsilon)}.
$$

Since $\| \cdot \|_{F_\beta(\vartheta)} \geq \| \cdot \|_{F(\vartheta)}$ for all $\beta > 0$ and $\vartheta \in \Theta$, we finally obtain that $k(\varepsilon)$ is also an upper bound for the covering number associated with Fisher balls, and this concludes the proof. $\qquad\square$

The following result exploits the link between the generalization bound and the covering bound proved in Lemma B.2.

**Lemma B.3.** *Under the assumption of Theorem 5.1, there exist $\varepsilon_0, C, K > 0$ such that for all $\varepsilon \in (0, \varepsilon_0)$ we have*

$$
\mathbf{P}\left\{ \sup_{\vartheta \in \Theta} |R(\vartheta) - R_n(\vartheta)| \geq C\varepsilon \right\} \leq 4\, k(\varepsilon) \exp\left( -Kn\varepsilon^{8/3} \right), \tag{25}
$$

*where $k(\varepsilon)$ is a bound on the cardinality of a covering by Fisher balls of radius $\varepsilon$.*

*Proof.* As a first step, we need to "discretize" the estimate of the left-hand side of (25) at the micro-scale $\varepsilon$, using the previous covering lemma (Lemma B.2). By inspecting the proof of Lemma B.2, we

notice that we can consider $\beta$-Fisher balls instead of Fisher balls for the covering, where $\beta$ is defined in (24). We also recall that the meso-scale $\delta$ is now given by

$$\delta = \left( \frac{\varepsilon}{\sqrt{d(3 + 5L\sqrt{d})}} \right)^{\zeta} .$$

Since $S_n(\vartheta) = R(\vartheta) - R_n(\vartheta)$, for all $\vartheta_1, \vartheta_2 \in \Theta$ we have

$$|S_n(\vartheta_1) - S_n(\vartheta_2)| \le |R(\vartheta_1) - R(\vartheta_2)| + |R_n(\vartheta_1) - R_n(\vartheta_2)| . \tag{26}$$

Now we estimate each term in the right-hand side of (26) under the assumption $|\vartheta_1 - \vartheta_2| < 5\sqrt{d}\delta$. We set $\vartheta(t) = t\vartheta_1 + (1 - t)\vartheta_2$ for $t \in [0, 1]$, and we estimate the first term:

$$
\begin{aligned}
|R(\vartheta_1) - R(\vartheta_2)| &\le \int_{\mathcal{X} \times \mathcal{Y}} |\mathfrak{L}(p_{\vartheta_1}(y|x)) - \mathfrak{L}(p_{\vartheta_2}(y|x))| \, p(dx, dy) \\
&\le \int_{\mathcal{X} \times \mathcal{Y}} \left| \mathfrak{L}(p_{\vartheta(0)}(y|x)) - \mathfrak{L}(p_{\vartheta(1)}(y|x)) \right| p(dx, dy) \\
&= \int_{\mathcal{X} \times \mathcal{Y}} \left| \int_0^1 \partial_1 \mathfrak{L}(p_{\vartheta(t)}(y|x), p(y|x)) \left\langle \nabla_\vartheta p_{\vartheta(t)}(y|x), \vartheta_2 - \vartheta_1 \right\rangle dt \right| p(dx, dy) \\
&\le \int_0^1 \int_{\mathcal{X} \times \mathcal{Y}} |\partial_1 \mathfrak{L}(p_{\vartheta(t)}(y|x), p(y|x))| \left| \left\langle \nabla_\vartheta p_{\vartheta(t)}(y|x), \vartheta_2 - \vartheta_1 \right\rangle \right| p(dx, dy) dt \\
&\le \Lambda \int_0^1 \int_{\mathcal{X} \times \mathcal{Y}} \left| \left\langle \nabla_\vartheta p_{\vartheta(t)}(x, y), \vartheta_2 - \vartheta_1 \right\rangle \right| p(dx, dy) \, dt \\
&= \Lambda \int_0^1 \int_{\mathcal{X} \times \mathcal{Y}} \left| \left\langle \nabla_\vartheta \log p_{\vartheta(t)}(x, y), \vartheta_2 - \vartheta_1 \right\rangle \right| p_{\vartheta(t)}(x, y) \, p(dx, dy) \, dt \\
&= \Lambda \int_0^1 \int_{\mathcal{X} \times \mathcal{Y}} \left| \left\langle \nabla_\vartheta \log p_{\vartheta(t)}(x, y), \vartheta_2 - \vartheta_1 \right\rangle \right| p(x, y) \, p_{\vartheta(t)}(dx, dy) \, dt ,
\end{aligned}
$$

where we have used the fundamental theorem of calculus, Fubini's theorem, the $\Lambda$-Lipschitzianity of $\mathcal{L}$, and the fact that $\nabla_\vartheta \log p_\vartheta(y|x) = \nabla_\vartheta \log p_\vartheta(x, y)$. Then, by Cauchy-Schwarz inequality, we obtain

$$
\begin{aligned}
|R(\vartheta_1) - R(\vartheta_2)| &\le \Lambda \int_0^1 \mathbb{E}_{p_{\vartheta(t)}}[p^2(x, y)]^{1/2} \cdot \left\langle F(\vartheta(t))(\vartheta_2 - \vartheta_1), \vartheta_2 - \vartheta_1 \right\rangle^{1/2} dt \\
&\le \Lambda \int_0^1 \mathbb{E}_{p_{\vartheta(t)}}[p^2(x, y)]^{1/2} \cdot \left\langle F_\beta(\vartheta(t))(\vartheta_2 - \vartheta_1), \vartheta_2 - \vartheta_1 \right\rangle^{1/2} dt \\
&\le \Lambda C_1 \int_0^1 \left\langle F_\beta(\vartheta(t))(\vartheta_2 - \vartheta_1), \vartheta_2 - \vartheta_1 \right\rangle^{1/2} dt ,
\end{aligned}
$$

for some constant $C_1 > 0$ depending on $\alpha_1, \alpha_2$, thanks to hypothesis (ii). Now, Lemma B.1 implies that

$$\left\langle F_\beta(\vartheta(t))(\vartheta_2 - \vartheta_1), \vartheta_2 - \vartheta_1 \right\rangle \le \left( 3 + \omega_\beta(t|\vartheta_2 - \vartheta_1|) \right) \|\vartheta_2 - \vartheta_1\|_{F_\beta(\vartheta_1)} . \tag{27}$$

By (27) and (23) we conclude that

$$
\begin{aligned}
|R(\vartheta_1) - R(\vartheta_2)| &\le \Lambda C_1 \int_0^1 \left( 3 + \omega_\beta(t|\vartheta_2 - \vartheta_1|) \right)^{1/2} dt \, \|\vartheta_2 - \vartheta_1\|_{F_\beta(\vartheta_1)} \\
&\le C_2 \|\vartheta_2 - \vartheta_1\|_{F_\beta(\vartheta_1)} ,
\end{aligned} \tag{28}
$$

where $C_2$ is a constant depending only on $\Lambda, C_1$ and the dimension $d$.

Now, by a similar computation, we estimate the second term in the r.h.s. of (26):

$$|R_n(\vartheta_1) - R_n(\vartheta_2)| \leq \Lambda \int_0^1 \left( \frac{1}{n} \sum_{i=1}^n \left\langle \nabla \log p_{\vartheta(t)}(X_i, Y_i), \vartheta_2 - \vartheta_1 \right\rangle^2 \frac{p_{\vartheta(t)}(X_i, Y_i)}{p(X_i, Y_i)} \right)^{1/2}$$

$$\cdot \left( \frac{1}{n} \sum_{i=1}^n p_{\vartheta(t)}(X_i, Y_i) p(X_i, Y_i) \right)^{1/2} dt$$

$$\leq \alpha_2 \int_0^1 \left( \frac{1}{n} \sum_{i=1}^n \left\langle \nabla \log p_{\vartheta(t)}(X_i, Y_i), \vartheta_2 - \vartheta_1 \right\rangle^2 \frac{p_{\vartheta(t)}(X_i, Y_i)}{p(X_i, Y_i)} \right)^{1/2} dt \,.$$

$$(29)$$

Let us set

$$Z_i(t) := \left\langle \nabla \log p_{\vartheta(t)}(X_i, Y_i), \vartheta_2 - \vartheta_1 \right\rangle^2 \frac{p_{\vartheta(t)}(X_i, Y_i)}{p(X_i, Y_i)}$$

and

$$T := \sup \frac{p_\vartheta(x, y)}{p(x, y)} |\nabla_\vartheta \log p_\vartheta(x, y)|^2 \,,$$

where the supremum is computed w.r.t $(x, y) \in \mathcal{X} \times \mathcal{Y}$ and $\vartheta \in \Theta$. By assumptions (i) and (ii) we obtain

$$T \leq B \sup |\nabla_\vartheta \log p_\vartheta(x, y)|^2 < \infty \,,$$

where $B = \alpha_2/\alpha_1$. Thus we also get

$$0 \leq Z_i(t) \leq T|\vartheta_2 - \vartheta_1|^2 \,.$$

The expectation of $Z_i(t)$ is

$$\overline{Z}_i(t) = \mathbb{E}_{(x,y)\sim p}[Z_i(t)] = \int \left\langle \nabla \log p_{\vartheta(t)}(x, y), \vartheta_2 - \vartheta_1 \right\rangle^2 \frac{p_{\vartheta(t)}(x, y)}{p(x, y)} p(dx, dy)$$

$$= \int \left\langle \nabla \log p_{\vartheta(t)}(x, y), \vartheta_2 - \vartheta_1 \right\rangle^2 p_{\vartheta(t)}(dx, dy)$$

$$= \left\langle F(\vartheta(t))(\vartheta_2 - \vartheta_1), \vartheta_2 - \vartheta_1 \right\rangle$$

hence also $\frac{1}{n} \sum_{i=1}^n Z_i(t)$ has the same expectation, by independence of the $Z_i(t)$.

Now, from Lemma B.2 we know that $\Theta$ can be covered with $k = k(\varepsilon) \leq C_d \varepsilon^{-d_\varsigma(\varepsilon)}$ $\beta$-Fisher balls $B_1, \ldots, B_k$ of size $\varepsilon$. Let now $\eta = C\varepsilon$ for some $C > 0$ to be chosen later, and evaluate

$$\mathbf{P}\left\{ \sup_{\vartheta \in \Theta} |S_n(\vartheta)| \geq \eta \right\} \leq \mathbf{P}\left\{ \bigcup_{j=1}^k \sup_{\vartheta \in B_j} |S_n(\vartheta)| \geq \eta \right\} \leq \sum_{j=1}^k \mathbf{P}\left\{ \sup_{\vartheta \in B_j} |S_n(\vartheta)| \geq \eta \right\} \,.$$

Now for all $j = 1, \ldots, k$ we bound the probability of an event involving the computation of the supremum of $|S_n(\vartheta)|$ over $B_j$ with another one involving only the pointwise evaluation of $S_n$ at the center $\vartheta_j$ of $B_j$. Indeed by (28) and (29), and with $\vartheta, \vartheta_j$ respectively replacing $\vartheta_2, \vartheta_1$, we deduce

$$\mathbf{P}\left\{ \sup_{\vartheta \in B_j} |S_n(\vartheta)| \geq \eta \right\}$$

$$\leq \mathbf{P}\left\{ |S_n(\vartheta_j)| + \sup_{\vartheta \in B_j} \left( |S_n(\vartheta) - S_n(\vartheta_j)| \right) \geq \eta \right\}$$

$$\leq \mathbf{P}\left\{ |S_n(\vartheta_j)| + \sup_{\vartheta \in B_j} \left( C_2 \|\vartheta - \vartheta_j\|_{F_\beta(\vartheta_j)} + \alpha_2 \int_0^1 \left( \frac{1}{n} \sum_{i=1}^n Z_i(t) \right)^{1/2} dt \right) \geq \eta \right\}$$

$$\leq \mathbf{P}\left\{ |S_n(\vartheta_j)| \geq \frac{\eta}{2} \right\} + \mathbf{P}\left\{ \exists t \in [0, 1] : \frac{1}{n} \sum_{i=1}^n Z_i(t) \geq \frac{\eta^2}{16\,\alpha_2^2} \right\} \,,$$

where in the last inequality we have used $\|\vartheta - \vartheta_j\|_{F_\beta(\vartheta_j)} < \varepsilon = \frac{\eta}{C}$ and required $C \geq 4C_2$. Owing to Lemma B.4 and (v), we get

$$\mathbf{P}\left(|S_n(\vartheta_j)| \geq \frac{\eta}{2}\right) = \mathbf{P}\left(|R_n(\vartheta_j) - R(\vartheta_j)| \geq \frac{\eta}{2}\right) \leq 2 \exp\left(-\frac{n\eta^2}{2b^2}\right). \tag{30}$$

and

$$\mathbf{P}\left\{\left|\frac{1}{n}\sum_{i=1}^{n} Z_i(t) - \|\vartheta - \vartheta_j\|_{F_\beta(\vartheta(t))}^2\right| \geq \xi\right\} \leq 2\exp\left(-\frac{2n\xi^2}{T^2|\vartheta - \vartheta_j|^2}\right). \tag{31}$$

By (31) we find

$$\mathbf{P}\left\{\exists t \in [0,1]: \frac{1}{n}\sum_{i=1}^{n} Z_i(t) \geq \frac{\eta^2}{16\,\alpha_2^2}\right\}$$

$$\leq \mathbf{P}\left\{\|\vartheta - \vartheta_j\|_{F_\beta(\vartheta(t))}^2 \geq \frac{\eta^2}{32\,\alpha_2^2}\right\}$$

$$+ \mathbf{P}\left\{\frac{1}{n}\sum_{i=1}^{n} Z_i(t) - \|\vartheta - \vartheta_j\|_{F_\beta(\vartheta(t))}^2 \geq \frac{\eta^2}{32\,\alpha_2^2}\right\}$$

$$\leq \mathbf{P}\left\{\|\vartheta - \vartheta_j\|_{F_\beta(\vartheta(t))}^2 \geq \frac{\eta^2}{32\,\alpha_2^2}\right\} + 2\exp\left(-\frac{n\eta^4}{2^9\,\alpha_2^4 T^2 |\vartheta - \vartheta_j|^2}\right)$$

$$\leq 2\exp\left(-C_4 n\eta^{4-2\zeta}\right), \tag{32}$$

where

$$C_4 = \frac{C_2^{2\zeta}}{3^2 2^{9-2\zeta}\alpha_2^4 T^2 d}$$

and where the last inequality follows from

$$\|\vartheta - \vartheta_j\|_{F_\beta(\vartheta(t))}^2 < \frac{\eta^2}{32\,\alpha_2^2}, \tag{33}$$

which can be enforced by a suitable choice of the constant $C$, as explained hereafter. Indeed, using Lemma B.1 and (23) we obtain

$$\|\vartheta - \vartheta_j\|_{F_\beta(\vartheta(t))}^2 \leq (3 + \omega_\beta(t|\vartheta - \vartheta_j|))\|\vartheta - \vartheta_j\|_{F_\beta(\vartheta_j)}^2$$

$$\leq (3 + \omega_\beta(t|\vartheta - \vartheta_j|))\,\varepsilon^2$$

$$\leq (3 + 5L\sqrt{d})\varepsilon^2 \leq \frac{3 + 5L\sqrt{d}}{C^2}\eta^2.$$

Therefore, if we choose $C$ such that

$$(3 + 5L\sqrt{d}) < \frac{C^2}{32\alpha_2^2},$$

we obtain (33), as wanted. The proof of (32) is now complete.

Finally, by (30) and (32), and observing that the second exponential is the leading term, we get

$$\mathbf{P}\left(\sup_{\vartheta \in \Theta} |S_n(\vartheta)| \geq \eta\right) \leq \sum_{i=1}^{k} \mathbf{P}\left(\sup_{\vartheta \in B_i} |S_n(\vartheta)| \geq \eta\right)$$

$$\leq 2\,k(\varepsilon)\left[\exp\left(-\frac{n\eta^2}{2b^2}\right) + \exp\left(-C_4 n\eta^{4-2\zeta}\right)\right]$$

$$\leq 4\,k(\varepsilon)\,\exp\left(-C_5 n\eta^{8/3}\right),$$

where $C_5 = \min(C_4, (2b^2)^{-1})$ and owing to $4 - 2\zeta < 3$, which follows from assumption (v). In conclusion we obtain (25) with $K = C_5 C^{8/3}$. □

*Proof of Theorem 5.1.* We choose $\gamma > 0$, and let $\varepsilon_n = \left(\frac{\log n}{\gamma n}\right)^{3/8}$ and $K$ be as in Lemma B.3. By combining Lemma B.2 with Lemma B.3 we obtain

$$\mathbf{P}\left(\sup_{\vartheta \in \Theta} |R(\vartheta) - R_n(\vartheta)| \geq C\varepsilon_n\right) \leq 4\, k(\varepsilon_n) \exp\left(-K\frac{\log n}{\gamma}\right)$$

$$\leq H\varepsilon_n^{-d_\zeta(\varepsilon_n)} n^{-\frac{K}{\gamma}},$$

where $C$ is as in Lemma B.3 and $H = 4C_d$. $\qquad\qquad\square$

We can now explain Remark 5.2 by noting that, if we choose

$$0 < \gamma < \gamma_0 := \frac{8K}{3d(1 + \log(1 + \mu^{1/2}))}, \tag{34}$$

with $K$ as in Lemma B.3, then the generalization bound becomes infinitesimal as $n \to +\infty$. Indeed, by the upper estimate (11) we have

$$d_\zeta(\varepsilon) \leq \zeta d + \hat{r}\left(1 - \zeta + \frac{\log(1 + \mu^{1/2})}{|\log(\varepsilon)|}\right) \leq d(1 + \log(1 + \mu^{1/2})) =: \bar{d},$$

whenever $\varepsilon < \exp(-1)$, so that

$$\varepsilon_n^{-d_\zeta(\varepsilon_n)} n^{-\frac{K}{\gamma}} = \left(\frac{\gamma n}{\log n}\right)^{3d_\zeta(\varepsilon_n)/8} n^{-\frac{K}{\gamma}} \leq \gamma^{3\bar{d}/8} n^{3\bar{d}/8 - K/\gamma}. \tag{35}$$

Hence, the infinitesimality of the generalization bound as $n \to \infty$ follows from $3\bar{d}/8 - K/\gamma < 0$, as wanted.

We recall Hoeffding's estimate, which is used in the proof of Lemma B.3.

**Lemma B.4** (Hoeffding's estimate). *Let $Z_i$, $i = 1, \ldots, n$, be independent random variables, such that $Z_i \in [a, b]$ almost surely. Define $V_n = \frac{1}{n}\sum_{i=1}^{n} Z_i$ and take $\varepsilon > 0$, then*

$$\mathbf{P}\left(|V_n - \mathbb{E}[V_n]| \geq \varepsilon\right) \leq 2\exp\left(-\frac{2n\varepsilon^2}{(b-a)^2}\right).$$

## C   Figures

The appendix contains a comprehensive collection of figures and tables that complement and enhance the understanding of the main content presented in this document. These figures provide visual representations of the results related to the experiments section discussed in the main part of the paper.

The results in Figure 4 and Figure 5 show that the impact of $\sigma^2$ on $\underline{d}_\zeta$ and $d_\zeta$ is negligible. To avoid discrepancies, we fix the data and the parameters used to estimate both $\underline{d}_\zeta$ and $d_\zeta$ via Monte Carlo integration. The meaningfulness of 2sED and lower 2sED for deterministic deep learning models is then enforced by taking the limit as $\sigma^2 \to 0$.

Table 2: Number of parameters of CNNs

| MODEL | NUMBER OF PARAMETERS ($d$) |
|---|---|
| CNN 7-5\|10-50-34-10 | 4493 |
| CNN 5-7\|10-50-34-10 | 4753 |
| CNN 5-4-3\|10-50-34-10 | 4985 |

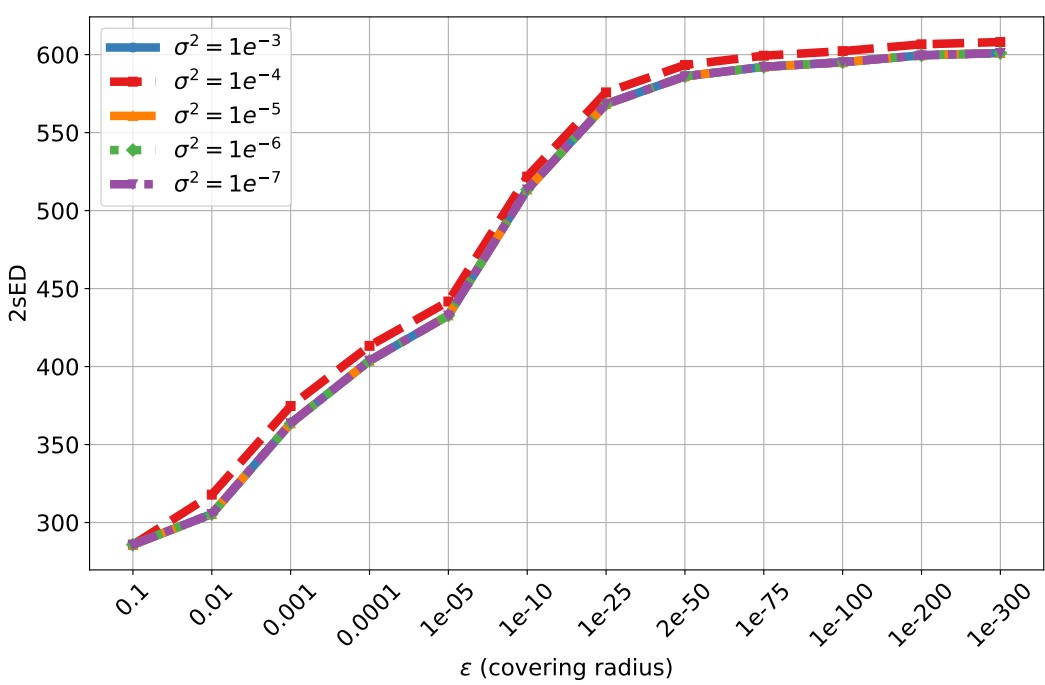

Figure 4: Impact of $\sigma^2$ on $d_\zeta$ for $MLP$ 54-16-7. The 2sED is estimated with a fixed seed varying $\sigma^2$.

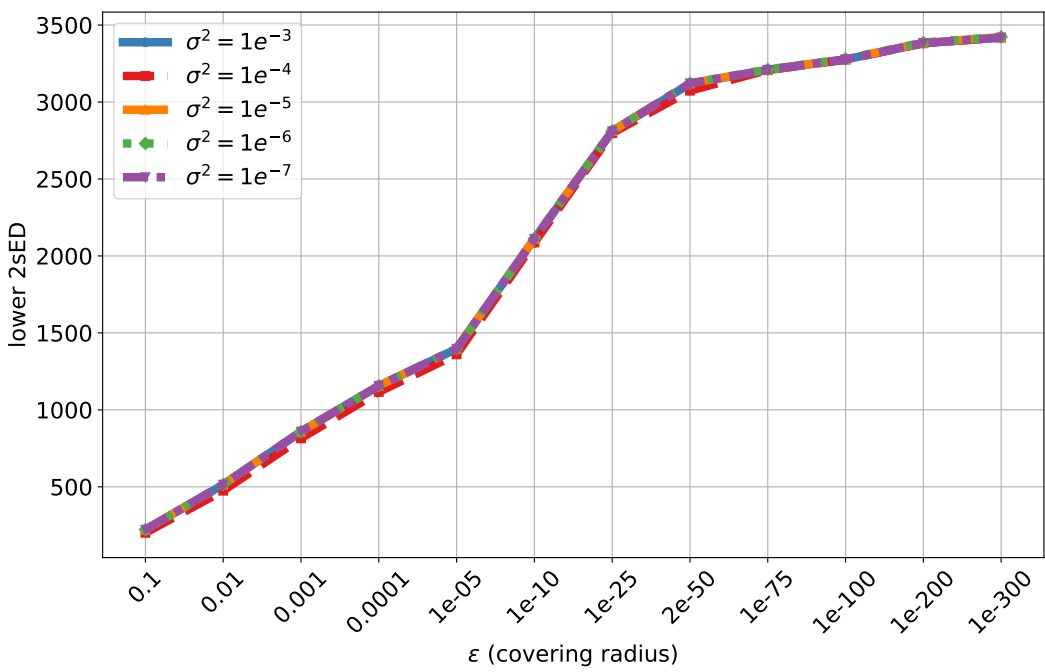

Figure 5: Impact of $\sigma^2$ on $d_\zeta$ for $CNN$ 5-7|10-50-34-10. The 2sED is estimated with a fixed seed varying $\sigma^2$

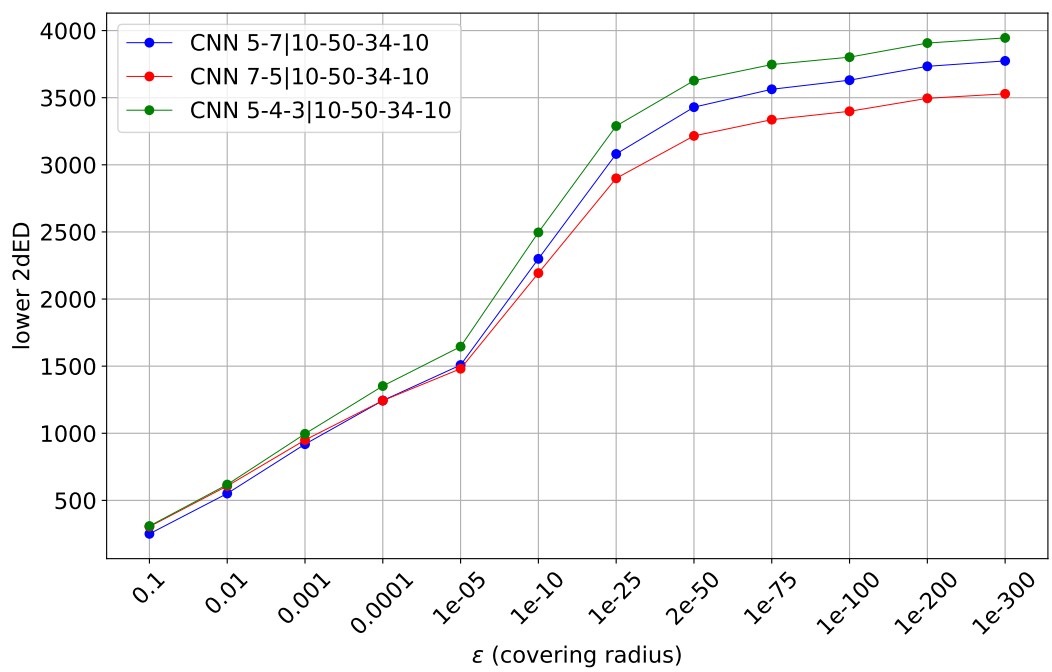

Figure 6: This figure plots the lower 2sED of CNNs estimated using 100 MNIST samples and 100 vectors of parameters for the Monte Carlo approximation.

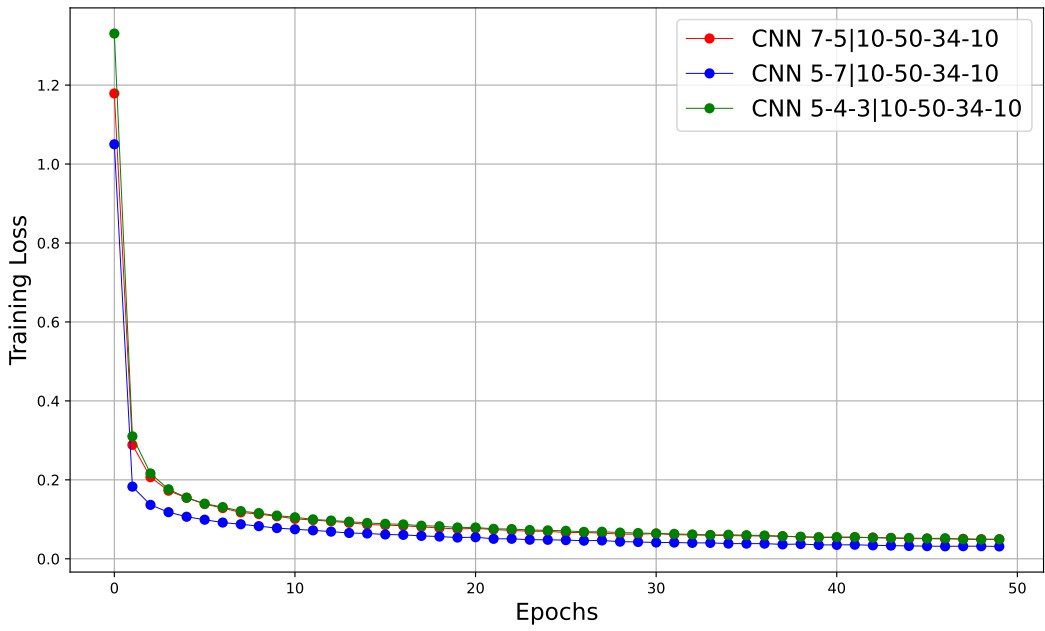

Figure 7: Training loss plots of CNNs on MNIST using Adam with learning rate $1e^{-3}$ and a batch size 256.

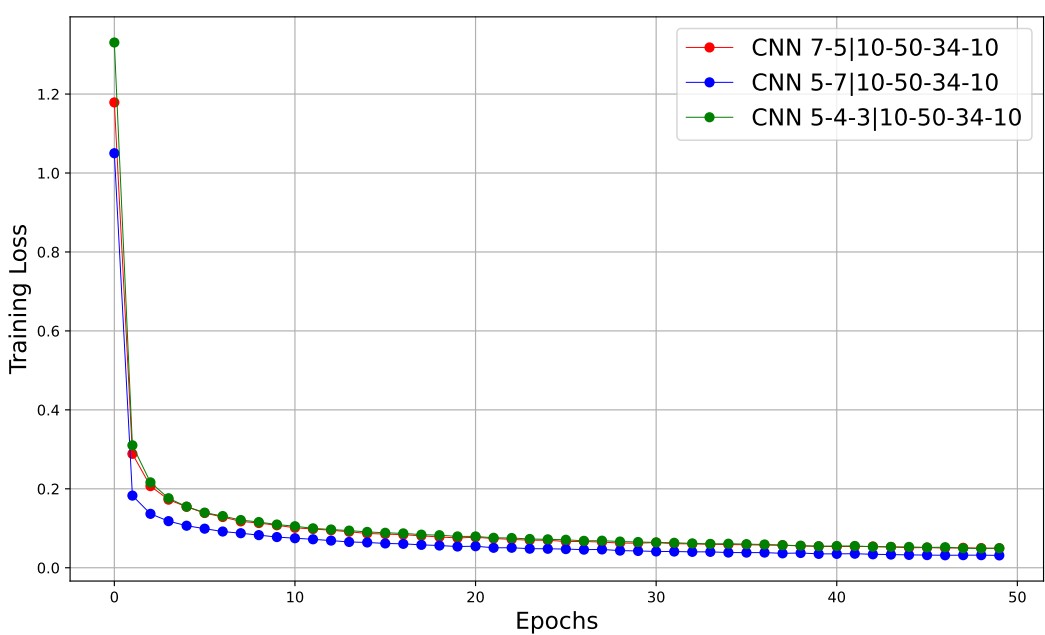

Figure 8: Training loss plots of CNNs on MNIST using Adam with learning rate $1e^{-3}$ and a batch size $512$.

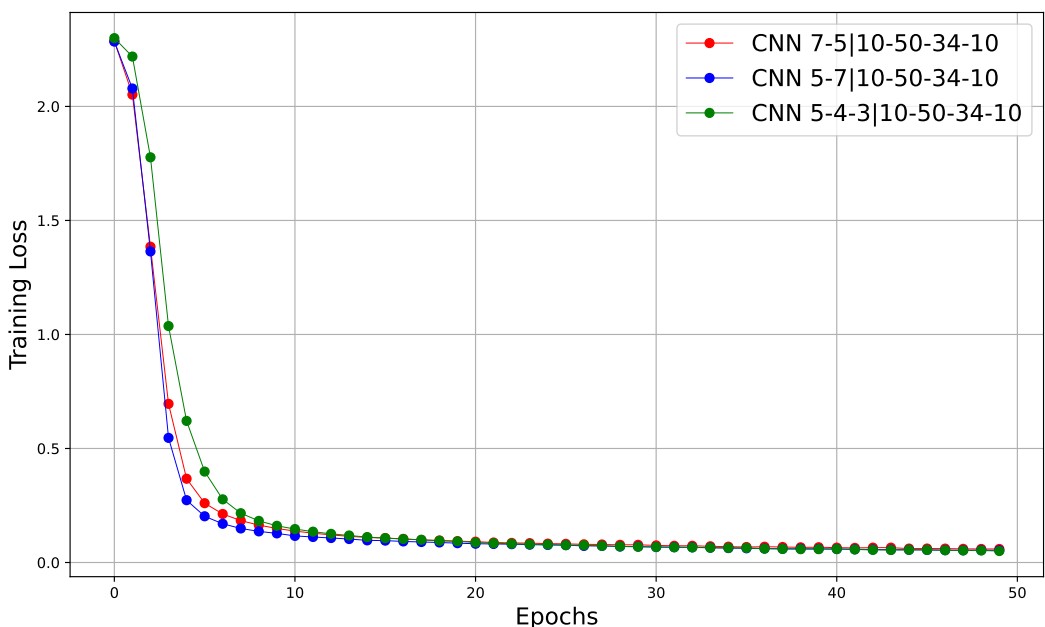

Figure 9: Training loss plots of CNNs on MNIST using Adam with learning rate $1e^{-3}$ and a batch size $2048$.

