# OpenReview forum: "A two-scale Complexity Measure for Deep Learning Models"
_NeurIPS.cc/2024/Conference — NeurIPS 2024 poster_

### Official Review · Reviewer_5Uzy · 2024-06-24

**Soundness:** 2
**Presentation:** 3
**Contribution:** 3
**Rating:** 5
**Confidence:** 4

**Summary:**

This paper proposes a new capacity measure for a general statistical model, called two-scale effective dimension (2sED), and provides an upper bound (under some assumptions on the model) on the generalisation error for this model based on the proposed measure. Then, the authors show how to lower bound the proposed 2sED for Markovian models such as feed-forward neural networks.

**Strengths:**

The authors obtain a nice upper bound on the generalisation error for any statistical model (including deep learning models) that satisfies some key conditions (please see my comment on these conditions below).

**Weaknesses:**

+ Theorem 5.1 is obtained thanks to some assumptions on the statistical model, some of them look very hard to verify for practical deep neural networks (DNNs).  For example, the assumption (iii) that the Fisher matrix $F(\upsilon)$ is $L$-Lipchitz looks too ideal, and I don't know there exists any DNNs such that this assumption holds.
+  Notations are not consistent: for example (2) and (19).

**Questions:**

+ Please explain why (iii) holds for some deep neural networks (DNNs)? Can you give an example of a DNN such that this assumption holds?
+ Please explain the last step in the proof of Lemma B.1.
+ What is $S_{\epsilon}$ in the line 456?
+ Please explain the last equality in p.15 (refer to the definition of $d_{\zeta}(\epsilon)$ in (4)).
+ The definition of $d_{\zeta}(\epsilon)$ in (9) looks not the same as (4) (for example, as $L=1$).

**Limitations:**

This is a theoretical research paper, hence the negative society impact of this work is not direct. The authors mention some technical limitations of this work in Section 7.

---

> ### Author Rebuttal · Authors · 2024-08-06
>
> We thank the reviewer for their time and the feedback.
> We next answer the criticism/questions:
>
>
> **(1) Weakness comment about the notation:**
>
> Thank you for the comment. We will make the notation consistent.
>
>
> **(2) Question regarding Assumption (iii)**
>
> We agree with the reviewer that assumption (iii) is quite strong. However, the proof of Theorem 5.1  can be easily changed using Fisher balls (more similar to the classical Euclidean balls) instead of the Fisher boxes assuming only the Lipschitz regularity of the FIM. This assumption is much weaker than assuming the Lipschitz regularity of the eigenvectors and it is satisfied by, for instance, FNN with a sigmoid activation function (or other smooth enough activation functions). We will update the paper with the new proof as soon as possible. We thank the reviewer again for the comment that helped us to improve our result.
>
>
> **(3) Question regarding the last step in Proof of Lemma B.1:**
>
> Here are additional explanations with the new balls:
> $$||v||^2_{A_{\beta}(\theta_1)}  = <A_{\beta}(\theta_1)v,v> = \sum_{i=1}^d \lambda_{i,\beta}(\theta_1)|<v,u_i(\theta_1)>|^2 \ge \beta |v|^2$$
>
>
> **(4) Question about S_{epsilon} in line 456:**
>
> Typos, it should be $S_Q$.
>
>
> **(5) Question about last equality in p.15:**
>
> This follows from writing the integral as $\epsilon$ to the $\log_{\epsilon}$ of the integral and changing the base of the logarithm from $\epsilon$ to $e$.
>
>
> **(6) Question regarding def (9) vs def (4):**
>
> For Markovian models, the two definitions coincide since the FIM factorizes and hence det(F) is the product of the determinant of each block. We write F instead of $\hat{F}$ just for convenience as explained in line 148 page 4.
>
>
> **Concluding comments:**
>
> We thank the reviewer for the constructive feedback again. We hope that our explanations above make clear that the assumptions of Thm. 5.1 are clearly there but are not as restrictive as they may seem at first sight (assumption (iii) can be weakened as said in the answer to question (2)). We also want to highlight that totally removing (or even relaxing) the assumption does not seem possible to us (without losing the generalization bound given by Thm 5.1).

---

> > ### Comment · Reviewer_5Uzy · 2024-08-14
> > **Reply to the authors' rebuttals**
> >
> > Thank you very much for your answers to my questions. Although the assumption (iii) looks too ideal, but I understand that getting rid of this condition is too challenging in a short time. Hence, I raised my score to 5.

---

### Official Review · Reviewer_MhtR · 2024-07-08

**Soundness:** 3
**Presentation:** 3
**Contribution:** 3
**Rating:** 6
**Confidence:** 3

**Summary:**

In this work, a new measure of model complexity, 2sED, is introduced. It is used to derive a new generalization bound for statistical models. A special case of 2sED is shown for Markovian models. Experiments show that the complexity measure correlates with the training loss of neural networks.

**Strengths:**

1.	Addresses an important and challenging topic.

2.	Solid mathematical results.

3.	Clearly written.

**Weaknesses:**

-	The main weakness is that the significance of the results are not clear. Concretely, it seems that when using the empirical Fisher matrix (as in the experiments), the generalization bound has the same value for all models that perfectly fit the training set. Therefore, it is not clear how this result can facilitate model selection. Furthermore, it is not clear how the bound can shed light on the generalization performance of neural networks, since 2sED is not simple to analyze.

-	Some technical details are missing. (1) The derivation of 2sED from the effective dimension (2) Explicit example of a Markovian model (e.g., a feed-forward network).

-	The empirical results are limited showing correlation only with the training loss and not the validation/test loss which is the main goal.

**Questions:**

-	Does the generalization bound have the same value for all networks that perfectly fit the training set, if the empirical Fisher matrix is used?
-	Does 2sED satisfy (P2) in line 19?

**Limitations:**

Not all limitations are mentioned (e.g. experiments only show correlation of 2sED with the training loss and not the validation/ test loss).

---

> ### Author Rebuttal · Authors · 2024-08-06
>
> We thank the reviewer for their time and the feedback.
>
> We next answer the questions/comments raised:
>
> **(1) Main weakness regarding the significance:**
>
> It is correct that 2sED is in general expensive to calculate. However, we show that for the case of Markov models we have a lower bound that can be computed efficiently and that performs well in practice (in the sense that it is close to 2sED for these models). We further want to emphasize that the 2sED may be computationally expensive to compute for a general model, but it is still much easier to compute than other complexity measures in general, like the VC dimension and the Rademacher complexity that can only be estimated.
>
>
>
> **(2) Comment regarding technical technical details missing:**
>
> We have described how feed-forward and convolutional neural networks can be seen as Markovian models in the experiment section. Furthermore, there is no way to pass from the effective dimension to the 2sED. It is another definition inspired by the same motivation.
> We are happy to add these details in an updated version of the manuscript.
>
>
>
> **(3) Comment regarding the limited empirical experiments:**
>
> Our empirical analysis of the correlation between the (lower) 2sED and the post-training performances of neural networks with different topologies but with a similar number of parameters is supported by various experiments conducted on qualitatively different datasets and networks. Moreover, one has to take into account that our work contains a consistent theoretical part, and the experiments are meant to illustrate some features and potentials of the (lower) 2sED in the context of Markov models/feedforward networks.
>
>
> **(4) Question “Does the generalization bound have the same value for all networks that perfectly fit the training set, if the empirical Fisher matrix is used?”**
>
> Even if the empirical FIM is used, having two models that perfectly fit the training set would mean that $1/n \sum_j \nabla_\theta \log p^i_\theta (x_j,y_j) = 0$ for $i=1,2$ where $p^i_\theta$ is the probability distribution associated to the two models respectively and $\theta$ is the optimal point. This does not imply that the empirical FIM of the two models is the same since the sum of the tensor product is not the tensor product of the sums. Indeed, the FIM and the empirical FIM evaluated in the optimal point are meant to give some information about the curvature of the log-likelihood around the optimal point.
>
> **(5) Question: “Does 2sED satisfy (P2) in line 19?”**
>
> Both the 2sED and the lower 2sED satisfy assumption (P2) in the case of the Markovian model. The main advantage of using the lower 2sED is that the bias introduced by a Monte Carlo estimation of the integral is reduced by the fact that, except for the first integral, the integral is outside the logarithm.
>
>
> **Concluding comments:**
>
> We thank the reviewer for the constructive feedback again. We hope that our explanations above were helpful in removing some criticism about the submitted manuscript.

---

> > ### Comment · Reviewer_MhtR · 2024-08-08
> >
> > Thanks, I see my mistake with the empirical FIM comment. I will raise the score.

---

> > > ### Author Response · Authors · 2024-08-08
> > >
> > > Thank you very much

---

### Official Review · Reviewer_aPDv · 2024-07-14

**Soundness:** 4
**Presentation:** 4
**Contribution:** 3
**Rating:** 7
**Confidence:** 3

**Summary:**

This paper proposes a two-scale complexity measure that can be used to derive generalisation bounds on the empirical risk. This measure is intuitive and comes from the box-counting dimension of the parameter space with the Fisher metric. From what I can understand, authors please correct me if I'm wrong, the idea is that under the Fisher geometry, this describes how complex the parameter space is. We are seeking to identify the intrinsic complexity ( or stand in, dimension ) of this space as a surrogate for how difficult our modelling problem is. It seems natural to me that then the effective dimension of parameter space would then control the generalisation error.

**Strengths:**

This is a very nicely written learning theory paper. I find the complexity measure defined very well motivated and the definitions given by the authors striking a good balance between mathematical precision and applicability to large deep learning architectures. The mathematical proofs are well presented and the correct amount of detail in the main text is given for a conference paper.

The true strength of this paper is the simplicity of what the authors are proposing as a complexity measure. I believe that the Fisher information is the correct Hilbert space to analyse such complexity and intrinsic dimensions are natural here. The theory is shown to correlate with simple experiments.

**Weaknesses:**

This paper fall folly to a hard problem with both fractal dimensions and Fisher information matrices. Computing the eigenvalues of the FIM is a computationally intensive task and hinders the current proposed method. Further, it is well known that box-counting dimensions can be extremely difficult to estimate, particularly when boxes may not cover the target set efficiently. Both of this computational bottlenecks prevent this work being directly applicable to analysis of deep learning models.

**Questions:**

My first question is does this work bare any similarity to the following papers:

A. Camuto, G. Deligiannidis, M. A. Erdogdu, M. Gurbuzbalaban, U. Simsekli, and L. Zhu, "Fractal structure and generalization properties of stochastic optimization algorithms", NeurIPS (Spotlight), vol. 34, 2021.

B. Dupuis, G. Deligiannidis, and U. Simsekli, "Generalization Bounds with Data-dependent Fractal Dimensions", ICML, 2023.

In particular, the latter paper employs techniques from TDA to look at a range of box-covering sizes and add stability to the proposed method in this way. Do the authors think that possibly making their complexity measure a barcode could help make the derived statistic more stable?

My second question moves to the practical aspect of the authors work. Can the authors put in a section on the approximation of eigenvalues for their method? It has been hinted toward in the OOD literature that the FIM of large neural networks has a sparse structure. There are eigenvalue methods for sparse matrices which I think to benefit the authors work and believe this is worth putting in the main text to assist the community with evaluating the method.

My last question is: box-counting dimensions are typically a surrogate for some form of Hausdorff dimension. Do the authors think that simple Fisher boxes are the best choice or could another way of covering the desired sets be more appropriate in practice. I also usually understand that we typically need lower and upper box-counting dimensions, can the bound derived in the authors main theorem be also compared with what we might expect with the lower/upper box counting dimensions if these are not equal?

**Limitations:**

The authors do an excellent job at stating their mathematical assumptions and computational bottlenecks. This paper honestly addresses its limitations.

---

> ### Author Rebuttal · Authors · 2024-08-06
>
> We thank the reviewer for their time and the positive feedback.
> Your summary of the paper is correct and very nice.
> Your  “weakness” comment is also correct. Computing the eigenvalues of FIM is computationally intensive (as the FIM get’s very large). One main contribution of our work is that we show in the case of Markov models we can compute the ED efficiently in an iterative fashion (layer-by-layer).
>
> **Answer to your Question 1 [regarding similarities to existing works]**
>
> We thank the reviewer for pointing out other work that can be potentially related to a more geometric notion of complexity.
> The first work studies more the generalization properties after training of a machine learning model using the box-counting dimension (with respect to the Euclidean metric) of the support of the invariant measure induced on the parameter set by the training process. Even if the box-counting dimension is computed with respect to the Euclidean metric and the model architecture arises only in the support of the invariant measure, it is an interesting research question to study the 2sED replacing the Lebesgue measure with the invariant measure induced by the training process.
> The second article is much more related with the 2sED. Their idea is to include some geometrical properties of the loss into the generalization bound, but their metric (see (8)) is different than the Fisher metric used in our work in general. The main difference is that the 2sED introduces a “fractal dimension” which is data-dependent, in the sense that the 2sED of a set of hypotheses varies based on the amount of training data at disposal. This connection is incorporated in the scale parameter appearing in the definition and we believe it is an important feature of the 2sED, since it seems to correlate well with the training loss. Also, the connection between the box-counting dimension and the persistent homology dimension holds at the limit of the covering radius (see (Kozma et al.,2005;) cited in the second article).
>
>
> **Answer to your Question 2 [regarding practical aspects]**
>
> We agree with the reviewers, in the final version we will add an extra section pointing out some methods for the eigenvalues approximation.
>
>
> **Answer to your Question 3 [regarding box counting dimension]**
>
> The choice of the Fisher boxes becomes natural if we consider the Riemannian metric induced by the FIM over the statistical manifold. In principle one could define other metrics, but the one induced by the FIM seems to be the most used and validated by the literature (see reference [30], [6], [23] ).
> The lower and upper box-counting dimensions are related with the existence of the limit in the definition of the Minkowski–Bouligand dimension. The 2sED is inspired by the box-counting dimension but the radius of the covering sets is decided in relation to the number of training samples (think of it as a resolution at which you look at the model). Therefore the limit on the covering radius is not present in the definition of the 2sED.

---

> > ### Comment · Reviewer_aPDv · 2024-08-09
> >
> > I thank the reviewers for their detailed response which has helped me be assured in my assessment. I believe this paper would be a good addition to the literature and stand by my rating of acceptance.

---

> > > ### Author Response · Authors · 2024-08-09
> > >
> > > Thank you.

---

### Official Review · Reviewer_iesa · 2024-07-16

**Soundness:** 3
**Presentation:** 3
**Contribution:** 3
**Rating:** 6
**Confidence:** 1

**Summary:**

The authors introduce a new complexity measure for machine learning models. The complexity measure induces generalization bounds and admits approximations for Markovian models. The authors show that this method is helpful for estimating the generalization error on several feed-forward neural networks.

**Strengths:**

- The authors study a fundamental problem in theoretical machine learning and introduce a novel approach to this problem
- Unlike many other complexity metrics, this one admits efficient approximations in many settings
- Empirically, the model seems to have predictive utility in terms of predicting model performance

**Weaknesses:**

- A more thorough discussion of previous work on the expressivity of neural networks (and how this method fits into that literature), would be helpful to better evaluate the strength of the contributions.

**Questions:**

How does the method fit into the existing literature on understanding the expressivity of neural networks?

**Limitations:**

I do not see any ethical and societal implications of the work that need to be discussed.

---

> ### Author Rebuttal · Authors · 2024-08-06
>
> We thank the reviewer for their time and the positive feedback.
> We are happy to add a more thorough discussion of previous work on the expressivity of neural networks to the manuscript.
> The effective dimension (ED) is a rather novel capacity measure that has been introduced recently (by Figalli et al.). Later a generalization bound has been proven (see https://www.nature.com/articles/s43588-021-00084-1) which justifies calling it a capacity measure. In https://arxiv.org/pdf/2112.04807 some of us presented an overview (see Table 1) how the ED relates to other (more standard) capacity measures.
> In the submitted work we further develop the ED by finding a solution to its main weakness, namely that it is hard to evaluate (like many other capacity measures too). We show that for Markov models we can approximate the ED efficiently.

---

> > ### Comment · Reviewer_iesa · 2024-08-12
> >
> > I acknowledge the author's response. I am inclined to keep my score.

---

### Decision · Program_Chairs · 2024-09-25

**Decision:**

Accept (poster)

**Comment:**

This paper studies complexity measures that can inform model selection and generalization bounds. It draws ideas from notions of dimensionality of data and proposes a two-scale effective dimension to characterize model complexity, leading to a generalization bound under suitable assumptions. Since computing this quantity is in general hard, the paper furthermore proposes a modular approach (which is in fact part of their desiderata of complexity measures in the case of compositional models) to estimate this quantity for Markovian models. The paper is also accompanied by experiments on feed-forward neural networks.

All reviewers agree in that the proposed results are interesting, the clarify of presentations is good, and that this work would be a valuable addition to the literature on learning theory - and I agree.